# REPLACEMENT LEARNING: TRAINING NEURAL NETWORKS WITH FEWER PARAMETERS

## ABSTRACT

Traditional End-to-End deep learning models typically enhance feature representation capabilities by increasing network depth and complexity. While such an approach improves performance, it inevitably leads to issues such as parameter redundancy and inefficient resource utilization, which become increasingly pronounced as the network deepens. Existing methods have attempted to alleviate these problems by skipping or removing redundant layers. However, they often rely on complex manual designs, which may result in performance degradation, increased computational costs, and reduced memory efficiency. To address these challenges, we propose a novel training paradigm termed Replacement Learning. This method selectively removes certain layers from the network and substitutes them with additional computing layers in an efficient and automated manner, thereby compensating for the potential performance loss caused by layer removal. Specifically, a computing layer is inserted between the neighboring layers of the removed layer, and it utilizes parameters from the adjacent layers to construct a transformed parameter representation through a simple and efficient learnable block. This transformed representation is then used to perform additional computation on the output of the preceding layer, yielding the final output passed to the subsequent layer. Furthermore, to accommodate architectural variations such as feature map sizes and channel dimensions in different network types, we design a tailored, lightweight learnable block accordingly. Replacement Learning leverages the contextual flow of information between adjacent layers to eliminate unnecessary computation, significantly reducing computational complexity, saving GPU memory usage, and accelerating training. More importantly, it achieves a balanced integration of historical context and newly introduced features, thereby enhancing the overall model performance. We validate the effectiveness of Replacement Learning on five benchmarks—CIFAR-10, STL-10, SVHN, ImageNet, and COCO—across classification and detection tasks using both CNNs and ViTs architectures. Results demonstrate that our method not only significantly reduces the number of network parameters, shortens training time, and lowers memory consumption, but also surpasses traditional End-to-End trained models in performance.

## 1 INTRODUCTION

Updating learnable parameters is fundamental for training deep learning models Yang et al. (2019). The most common method, global backpropagation Mostafa et al. (2018), is widely applied in fields like computer vision Yoo (2015); Voulodimos et al. (2018), natural language processing Goldberg (2016; 2017), and speech processing Ahmad et al. (2004); Chauvin & Rumelhart (2013). However, increasing model capabilities inevitably raise network depth and complexity, sharply escalating the computational and parameter demands of global backpropagation Nawi et al. (2008), which challenges GPU processing power and memory capacity Bragagnolo et al. (2022). Moreover, high similarity in learning patterns between neighbouring layers Kleinman et al. (2021) causes parameter redundancy and inefficient resource usage. With large models becoming prevalent, developing effective training methods to reduce computation time and save GPU memory while preserving performance is urgently needed.

To tackle the challenges of traditional backpropagation (BP) Mostafa et al. (2018), researchers have explored alternatives such as feedback alignment Lillicrap et al. (2014); Nøkland (2016, forward

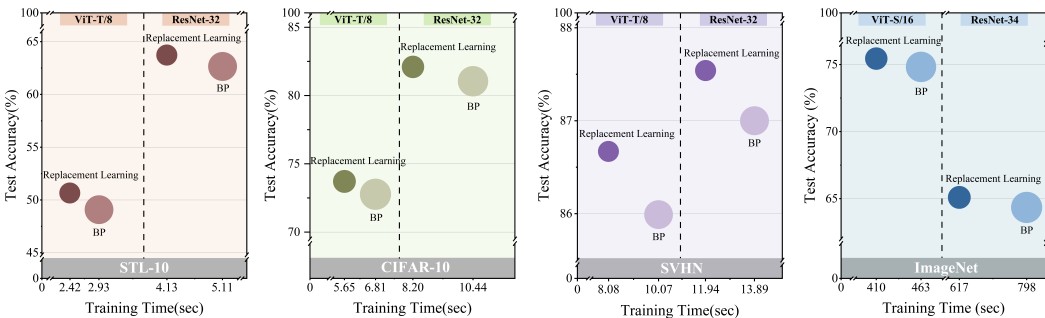

Figure 1: Comparison between different backbones with Replacement Learning and End-to-End training regarding GPU memory and Test accuracy. The diameter of the symbol is obtained based on GPU Memory at the same scale.

gradient learning Dellaferrera & Kreiman (2022); Ren et al. (2022), and local learning Su et al. (2024a;b). These methods aim to update network weights without fully relying on BP Rumelhart et al. (1985), thereby reducing training costs. However, they each have limitations. Feedback alignment struggles with training effectiveness due to inaccurate gradient estimation. Forward gradient learning requires extra forward passes, increasing computational overhead. Local learning divides the network into independently trained modules, but this often leads to suboptimal local performance and longer training times. Recent work on Vision Transformers (ViTs) Dosovitskiy et al. (2021) revealed strong inter-layer correlations from self-attention, leading to the skip attention Venkataramanan et al. (2023) approach to reduce complexity by reusing attention computations. However, this method requires manually designed auxiliary modules, making it complex and hard to generalize. Additionally, it risks error propagation, negatively impacting model performance. As a result, alternatives to backpropagation Rumelhart et al. (1985) and skip attention Venkataramanan et al. (2023) still face challenges in balancing training efficiency and computational cost while maintaining performance.

In this paper, we propose a novel method: Replacement Learning, which aims to significantly reduce the computational overhead and resource consumption of deep neural networks while maintaining—or even improving—model performance. The core idea of Replacement Learning is to selectively remove specific layers of the network and replace them with a lightweight computing layer that features a simple structure and minimal parameter count. Specifically, the computing layer synthesizes new computational parameters by integrating information from the parameters of the layers immediately preceding and succeeding the removed layer. This integration is accomplished through a specially designed, lightweight, learnable block. The fused parameters are then used to reprocess the output of the preceding layer, which is subsequently fed into the succeeding layer. This mechanism effectively compensates for the potential feature loss resulting from layer removal. The design notably enhances the network's capacity to capture local features in shallow layers and global representations in deeper layers, thereby promoting a more effective integration of low-level and high-level features. Moreover, we introduce an optimized interval strategy to regulate the frequency at which layers are removed and optimized, striking a desirable balance between computational efficiency and model performance. By leveraging two specially designed learnable blocks within the computing layer, Replacement Learning achieves efficient fusion of adjacent layer information and dynamically balances the retention of historical context with the incorporation of new feature representations, thereby further boosting overall performance. We comprehensively evaluate the effectiveness of Replacement Learning on five widely used benchmark datasets-CIFAR-10 Krizhevsky et al. (2009), STL-10 Coates et al. (2011), SVHN Netzer et al. (2011), ImageNet Deng et al. (2009), and COCO Lin et al. (2015)—across image classification and object detection tasks, employing both CNNs and ViTs Dosovitskiy et al. (2021) architectures. Experimental results demonstrate that, compared with traditional End-to-End training methods Rumelhart et al. (1985), Replacement Learning not only significantly reduces the number of trainable parameters, training time, and GPU memory usage, but also achieves superior performance in terms of model accuracy.

We summarize our contributions as follows:

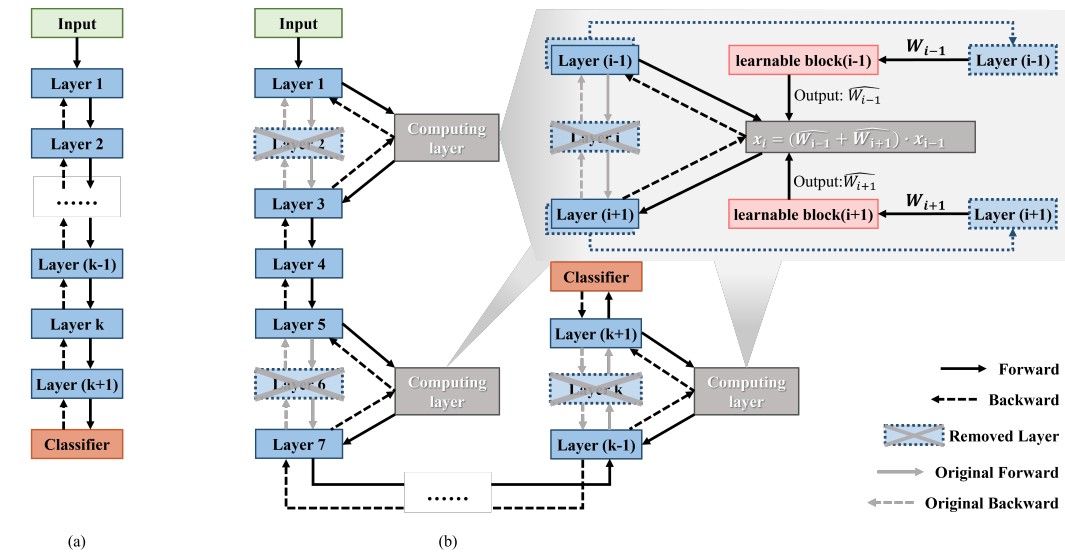

Figure 2: Comparison of (a) End-to-End training and (b) our proposed Replacement Learning.

- We propose a novel and general training method, Replacement Learning, which achieves performance comparable to or even surpassing that of traditional End-to-End training methods Rumelhart et al. (1985), while significantly reducing the number of parameters, training time, and GPU memory consumption.

- Replacement Learning is architecture and task-agnostic, exhibiting strong generalizability. It can be flexibly applied to models of varying depths and across different domains.

- We conduct extensive experiments on several widely-used image classification and object detection benchmarks, including CIFAR-10 Krizhevsky et al. (2009), STL-10 Coates et al. (2011), SVHN Netzer et al. (2011), ImageNet Deng et al. (2009), and COCO Lin et al. (2015). Results demonstrate that Replacement Learning consistently outperforms traditional End-to-End training methods in both computational efficiency and model performance.

## 2 METHOD

We present **Replacement Learning (RepL)**, which replaces every $k$-th block in a deep model with a lightweight learnable block that synthesizes an operator from the two neighbors' parameters and applies it in place of the removed block. This section specifies the exact implementation we use in our experiments for CNNs and ViTs: shapes, synthesis, forward computation, backward propagation.

### 2.1 PREPARATIONS

Let the network have depth $n$ and input $\mathbf{x}$; after operation $j$ the activation is $\mathbf{h}_j$ ($\mathbf{h}_0 = \mathbf{x}$). The standard forward is

$$\mathbf{h}_j = f_j(\mathbf{h}_{j-1}; \mathbf{W}_j), \qquad j = 1, \ldots, n, \tag{1}$$

where $f_j$ is a convolutional or transformer block with learnable weights $\mathbf{W}_j$. We replace every $k$-th site (except the last if $n$ is a multiple of $k$):

$$\mathcal{F} = \{ i \,|\, i \bmod k = 0, \ i < n \}. \tag{2}$$

For $i \in \mathcal{F}$, $f_i$ is not executed. Instead we run a learnable block that synthesizes an operator from $\mathbf{W}_{i-1}$ and $\mathbf{W}_{i+1}$ and applies it to $\mathbf{h}_{i-1}$, with normalization and nonlinearity preserved to match the baseline.

## 2.2 CNN LEARNABLE BLOCK

**Shapes.** At a replaced site $i \in \mathcal{F}$, the incoming feature is $\mathbf{h}_{i-1} \in \mathbb{R}^{C_{i-1}^{\text{in}} \times H \times W}$. The neighbor kernels (same $k \times k$ and stride in our settings) are

$$W_{i-1} \in \mathbb{R}^{C_{i-1}^{\text{out}} \times C_{i-1}^{\text{in}} \times k \times k}, \quad W_{i+1} \in \mathbb{R}^{C_{i+1}^{\text{out}} \times C_{i+1}^{\text{in}} \times k \times k}, \tag{3}$$

and the next site expects $C_{i+1}^{\text{in}}$ input channels.

**Synthesis via channel-mode learnable blocks.** We introduce two tiny **learnable blocks** acting on kernel channel modes:

$$\mathcal{T}_{i-1} : \ \mathbb{R}^{C_{i-1}^{\text{out}} \times C_{i-1}^{\text{in}} \times k \times k} \rightarrow \mathbb{R}^{C_{i+1}^{\text{in}} \times C_{i-1}^{\text{in}} \times 1 \times 1}, \tag{4}$$

$$\mathcal{T}_{i+1} : \ \mathbb{R}^{C_{i+1}^{\text{out}} \times C_{i+1}^{\text{in}} \times k \times k} \rightarrow \mathbb{R}^{C_{i+1}^{\text{in}} \times C_{i+1}^{\text{in}} \times 1 \times 1}. \tag{5}$$

Implementation: grouped $1 \times 1$ channel mixers (depth-wise $1 \times 1$), i.e., per-output-channel affine maps on the kernel tensor; parameter counts are only $C_{i-1}^{\text{out}}$ and $C_{i+1}^{\text{out}}$, respectively.

We fuse the aligned kernels into a valid $1 \times 1$ operator:

$$\widehat{W}_i \ = \ \mathcal{T}_{i-1}(W_{i-1}) \ + \ \mathcal{T}_{i+1}(W_{i+1}) \ \in \ \mathbb{R}^{C_{i+1}^{\text{in}} \times C_{i-1}^{\text{in}} \times 1 \times 1}. \tag{6}$$

**Forward.** The learnable block applies the synthesized operator and then matches the baseline nonlinearity/topology (BN + ReLU in our CNNs):

$$\hat{\mathbf{x}}_i \ = \ \widehat{W}_i * \mathbf{h}_{i-1}, \qquad \mathbf{h}_i \ = \ \phi\big(\text{BN}(\hat{\mathbf{x}}_i)\big). \tag{7}$$

Note: Eq. equation 6 is the linear part; the block mapping itself is nonlinear due to BN and ReLU.

**Backward.** Let the error arriving at $\hat{\mathbf{x}}_i$ be $\boldsymbol{\delta}_i$ and $G_i := \boldsymbol{\delta}_i \, \mathbf{h}_{i-1}^{\top}$ (channel-wise outer product). Then the learnable blocks receive gradients

$$\frac{\partial \mathcal{L}}{\partial \phi_{i-1}} = \langle G_i, W_{i-1} \rangle_{\text{channel}}, \qquad \frac{\partial \mathcal{L}}{\partial \phi_{i+1}} = \langle G_i, W_{i+1} \rangle_{\text{channel}}, \tag{8}$$

and the neighbor kernels get $\phi \odot G_i$ in addition to their own.

## 2.3 ViT LEARNABLE BLOCK

**Which weights are used.** All transformer submodule linears act in $\mathbb{R}^{d \times d}$. From the previous block, we collapse attention linears (Q/K/V and $W_o$) into $A_{i-1} \in \mathbb{R}^{d \times d}$ and the MLP linears into $M_{i-1} \in \mathbb{R}^{d \times d}$; similarly obtain $A_{i+1}, M_{i+1}$ from the next block.[1]

**Synthesis via learnable blocks implemented as parameters.** For ViTs, the learnable block is implemented as a pair of learnable parameters per fused operator:

$$\widehat{A}_i = \alpha_i A_{i-1} + \beta_i A_{i+1}, \qquad \widehat{M}_i = \alpha_i M_{i-1} + \beta_i M_{i+1}, \tag{9}$$

with $\alpha_i, \beta_i \in \mathbb{R}$ trained jointly with the model.

**Forward.** We apply two $d \times d$ linear transforms with LN + GELU and residual kept (as in our code and experiments):

$$\mathbf{H}_i \ = \ \text{LN}\Big(\text{GELU}\big(\widehat{M}_i \, \mathbf{H}_{i-1}\big) + \widehat{A}_i \, \mathbf{H}_{i-1}\Big) + \mathbf{H}_{i-1}. \tag{10}$$

**Backward.** Let $G_i := \boldsymbol{\delta}_i \, \mathbf{H}_{i-1}^{\top}$ at the two linear sites. Then

$$\frac{\partial \mathcal{L}}{\partial \alpha_i} = \langle G_i, A_{i-1} \rangle + \langle G_i, M_{i-1} \rangle, \qquad \frac{\partial \mathcal{L}}{\partial \beta_i} = \langle G_i, A_{i+1} \rangle + \langle G_i, M_{i+1} \rangle, \tag{11}$$

and neighbor weights receive $\alpha_i G_i$ and $\beta_i G_i$ contributions.

---

[1]Residual connections and LayerNorm remain outside and are kept.

## 2.4 Global forward with learnable blocks

The network with RepL executes

$$
\mathbf{h}_j = \begin{cases} f_j(\mathbf{h}_{j-1}; \mathbf{W}_j), & j \notin \mathcal{F}, \\ \phi\big(\mathrm{BN}(\widehat{W}_j(\mathbf{h}_{j-1}))\big), & j \in \mathcal{F} \ (\text{CNN}), \\ \mathrm{LN}\big(\mathrm{GELU}\big(\widehat{M}_j \mathbf{h}_{j-1}\big) + \widehat{A}_j \mathbf{h}_{j-1}\big) + \mathbf{h}_{j-1}, & j \in \mathcal{F} \ (\text{ViT}). \end{cases} \tag{12}
$$

with $\widehat{W}_j$ from Eq. equation 6 and $(\widehat{A}_j, \widehat{M}_j)$ from Eq. equation 9.

## 2.5 Operator ledger

**CNNs.**

- **Removed**: two $k \times k$ convs at depth $i$ and their intermediate BN activations.
- **Added**: two channel-mode $1 \times 1$ *learnable blocks* in weight space that synthesize $\widehat{W}_i$, and one BN+ReLU site to match topology.
- **Run-time effect**: conv MACs at site $i$ change from two $k \times k$ to two $1 \times 1$ applications; saved activations at this depth decrease accordingly.

**ViTs.**

- **Removed**: attention path (Q/K/V projections, $W_o$) and MLP ($d \to 4d \to d$).
- **Added**: two $d \times d$ linears built by a learnable block (parameters $\alpha_i, \beta_i$), with LN + GELU and residual kept.
- **Run-time effect**: arithmetic and saved activations at site $i$ drop to those of two $d \times d$ linear sites.

## 3 Experiments

### 3.1 Experimental setup

We conduct classification and detection experiments using different architectures on five benchmark datasets: CIFAR-10 Krizhevsky et al. (2009), STL-10 Coates et al. (2011), SVHN Netzer et al. (2011), ImageNet Deng et al. (2009), and COCO Lin et al. (2015).

During the experiment, we do not utilize pre-trained models. Instead, we train from scratch. We set $k = 4$ as the interval for the removed layer. All layers compute the loss using gradient descent and update the parameters via backpropagation Rumelhart et al. (1985).

### 3.2 Comparison with the E2E results

#### 3.2.1 Results on CIFAR-10, SVHN, and STL-10

We evaluate our method on CIFAR-10 Krizhevsky et al. (2009), SVHN Netzer et al. (2011), and STL-10 Coates et al. (2011), with results in Table 1. Replacement Learning (RepL) consistently outperforms End-to-End training Rumelhart et al. (1985) across all architectures: On CIFAR-10 Krizhevsky et al. (2009), ResNet-32/110 He et al. (2016) test accuracy rises from 93.17 to 93.43 and 93.49 to 94.01, while ViT-Tiny/8 Dosovitskiy et al. (2021) gains 0.94; on SVHN Netzer et al. (2011), accuracy increases by 0.13 at least across networks; on STL-10 Coates et al. (2011), gains range from 0.52 to 1.58, with consistent significant improvements across datasets. Table 1 also shows RL's advantages on CIFAR-10 Krizhevsky et al. (2009): ResNet-32/110 He et al. (2016) and ViT-Tiny/8 Dosovitskiy et al. (2021) reduce GPU memory by 0.69/1.69/0.73 GB, and training time per epoch by 21.5%, 20.1%, 17.0% respectively. Similar trends hold for SVHN Netzer et al. (2011) and STL-10 Coates et al. (2011), where RL cuts memory and training time while maintaining or improving performance.

Furthermore, when compared to Skip-Attention Venkataramanan et al. (2023) on ViTs Dosovitskiy et al. (2021), our method outperforms both in terms of performance and resource efficiency, making it a more favorable choice for maintaining accuracy while reducing computational cost.

Table 1: Performance of different backbones on various datasets. RepL represents Replacement Learning. Training time is the average result of each epoch.

| Dataset | Backbone | Method | Test Accuracy (%) | GPU Memory (GB) | Training Time (sec) |
|---|---|---|---|---|---|
| CIFAR-10 | ResNet-32 | E2E | 93.17±0.14 | 3.38 | 10.44 |
| | | RepL | 93.43±0.19 (↑0.26) | 2.69 (↓20.4%) | 8.20 (↓21.5%) |
| | ResNet-110 | E2E | 93.49±0.29 | 9.31 | 26.19 |
| | | RepL | 94.01±0.17 (↑0.52) | 7.62 (↓18.2%) | 20.93 (↓20.1%) |
| | ViT-Tiny/8 | E2E | 72.77±1.31 | 2.81 | 6.81 |
| | | Skip-Attention | 72.60±3.57(↓0.17) | 2.12(↓24.6%) | 6.23(↓8.5%) |
| | | RepL | 73.71±1.08 (↑0.94) | 2.08 (↓26.0%) | 5.65 (↓17.0%) |
| SVHN | ResNet-32 | E2E | 96.83±0.15 | 3.38 | 13.89 |
| | | RepL | 96.97±0.12 (↑0.14) | 2.69 (↓20.4%) | 11.94 (↓14.0%) |
| | ResNet-110 | E2E | 96.93±0.24 | 9.31 | 37.38 |
| | | RepL | 97.06±0.27 (↑0.13) | 7.62 (↓18.2%) | 30.08 (↓19.5%) |
| | ViT-Tiny/8 | E2E | 85.99±0.71 | 2.81 | 10.07 |
| | | Skip-Attention | 86.22±1.51(↑0.23) | 2.12(↓24.6%) | 9.18(↓8.8%) |
| | | RepL | 86.67±1.18 (↑0.68) | 2.08 (↓26.0%) | 8.08 (↓19.8%) |
| STL-10 | ResNet-32 | E2E | 79.81±0.51 | 3.38 | 5.11 |
| | | RepL | 80.33±0.42 (↑0.52) | 2.69 (↓20.4%) | 4.13 (↓19.2%) |
| | ResNet-110 | E2E | 79.78±0.30 | 9.31 | 6.86 |
| | | RepL | 80.45±0.51 (↑0.67) | 7.62 (↓18.2%) | 5.23 (↓23.8%) |
| | ViT-Tiny/8 | E2E | 49.08±3.39 | 2.81 | 2.93 |
| | | Skip-Attention | 50.42±3.18(↑1.34) | 2.12(↓24.6%) | 2.68(↓8.5%) |
| | | RepL | 50.66±3.18 (↑1.58) | 2.08 (↓26.0%) | 2.41 (↓17.8%) |

Table 2: Results on the ImageNet validation set. RepL stands for Replacement Learning. Training time is the average result of each epoch.

| Backbone | Method | Top-1 Accuracy (%) | Top-5 Accuracy (%) | GPU Memory (GB) | Training Time (sec) |
|---|---|---|---|---|---|
| ResNet-34 | E2E | 74.82±1.43 | 91.04±1.33 | 9.21 | 463.23 |
| | RepL | 75.44±1.27 (↑0.62) | 91.47±2.01 (↑0.43) | 8.06 ( ↓12.5% ) | 410.53 ( ↓11.4% ) |
| ResNet-101 | E2E | 77.55±1.22 | 93.80±1.78 | 20.95 | 720.11 |
| | RepL | 78.13±1.65 (↑0.58) | 94.02±1.34 (↑0.22) | 18.05 ( ↓13.8% ) | 616.23 ( ↓14.4% ) |
| ResNet-152 | E2E | 78.16±1.56 | 94.03±1.25 | 27.58 | 738.74 |
| | RepL | 78.31±1.46 (↑0.15) | 94.14±1.14 (↑0.11) | 24.19 ( ↓12.3% ) | 633.89 ( ↓14.2% ) |
| ViT-T/16 | E2E | 60.23±1.52 | 82.38±1.32 | 12.17 | 357.66 |
| | Skip-Attn | 60.51±1.20(↑0.28) | 82.72±1.09(↑0.34) | 11.52 ( ↓5.3% ) | 381.44 ( ↑6.7% ) |
| | RepL | 60.93±1.19 (↑0.70) | 82.88±1.07 (↑0.50) | 9.59 ( ↓21.2% ) | 290.15 ( ↓18.9% ) |
| ViT-S/16 | E2E | 64.35±1.83 | 84.64±1.22 | 21.05 | 798.61 |
| | Skip-Attn | 61.65±1.25(↓2.70) | 82.70±1.16(↓1.94) | 20.67 ( ↓1.8% ) | 755.14 ( ↓5.4% ) |
| | RepL | 65.09±1.41 (↑0.74) | 85.42±1.73 (↑0.78) | 16.22 ( ↓22.9% ) | 617.10 ( ↓22.7% ) |
| ViT-B/16 | E2E | 59.46±1.72 | 80.35±1.12 | 41.97 | 2566.70 |
| | Skip-Attn | 58.94±1.25(↓0.52) | 79.70±0.94(↓0.65) | 38.49 ( ↓8.3% ) | 2393.81 ( ↓6.7% ) |
| | RepL | 60.18±1.27 (↑0.72) | 81.97±1.15 (↑1.62) | 29.94 ( ↓28.7% ) | 1924.35 ( ↓25.1% ) |

### 3.2.2 RESULTS ON IMAGENET

We validate RepL's effectiveness on ImageNet Deng et al. (2009) with ResNet-34/101/152 He et al. (2016) and ViT-Tiny/16, ViT-Small/16, and ViT-Base/16 Dosovitskiy et al. (2021), and the results are shown in Table 2. For ResNet-34 He et al. (2016), Top-1 Accuracy rises from 74.82 to 75.44 and Top-5 from 91.04 to 91.47; the other five architectures also gain accuracy: Top-1 increases by 0.58, 0.15, 0.70, 0.74, 0.72 respectively, and Top-5 by 0.22, 0.11, 0.50, 0.78, 1.62 respectively.

Beyond accuracy, RepL reduces GPU memory usage and shortens per-epoch training time by 10%-25% across all models, highlighting its effectiveness on large-scale ImageNet Deng et al. (2009) even for deeper networks. Similarly, experiments on ViTs Dosovitskiy et al. (2021) with large datasets confirm our method outperforms the existing Skip-Attention Venkataramanan et al. (2023) mechanism.

## 3.3 ABLATION STUDY

### 3.3.1 PERFORMANCE ANALYSIS OF COMPUTING LAYER USAGE

To demonstrate the necessity of removing certain layers and the role of the computing layer as a replacement, we conduct comparative experiments on the CIFAR-10 Krizhevsky et al. (2009) using ViT-Tiny/8 Dosovitskiy et al. (2021) and ResNet-110 He et al. (2016). The performance of the traditional E2E training Rumelhart et al. (1985), a network with one-quarter of its layers removed according to the design with $k = 4$, and the network with the insertion of computing layers was evaluated and compared.

As shown in Table 3 and Table 4, after removing 25% of the layers, there is a significant reduction in GPU memory usage, and the training time is also considerably shortened. This demonstrates the positive impact of layer removal in terms of resource savings and efficiency enhancement. However, this comes at the cost of a decrease in accuracy. To address this limitation, we designed the insertion of computing layers in Replacement Learning to replace the removed layers. The results clearly indicate that our design is effective, as it not only saves GPU memory and reduces training time but also improves accuracy.

Table 3: Performance comparison on CIFAR-10.

| Backbone | Method | Test Accuracy (%) | GPU Memory (GB) | Training Time (sec) |
|---|---|---|---|---|
| ResNet-110 | E2E | 83.21±1.29 | 9.31 | 26.19 |
| | - 25% layers | 82.02±2.01 | 7.07 | 19.54 |
| | + computing layers | 83.95±1.17 | 7.62 | 20.93 |
| ViT-Tiny/8 | E2E | 72.77±1.31 | 2.81 | 6.81 |
| | - 25% layers | 71.13±1.24 | 2.04 | 5.44 |
| | + computing layers | 73.71±1.08 | 2.08 | 5.65 |

Table 4: Performance comparison on ImageNet.

| Backbone | Method | Top-1 Accuracy (%) | Top-5 Accuracy (%) | GPU Memory (GB) | Training Time (sec) |
|---|---|---|---|---|---|
| ResNet-34 | E2E | 74.82±1.43 | 91.04±1.33 | 9.21 | 463.23 |
| | - 25% layers | 72.99±1.82 | 90.12±1.31 | 7.75 | 392.21 |
| | + computing layers | 75.44±1.27 | 91.47±2.01 | 8.06 | 410.53 |
| ViT-Tiny/16 | E2E | 60.23±1.52 | 82.38±1.32 | 12.17 | 357.66 |
| | - 25% layers | 58.22±0.91 | 81.51±1.22 | 9.49 | 287.55 |
| | + computing layers | 60.93±1.19 | 82.88±1.07 | 9.59 | 290.15 |

### 3.3.2 ANALYSIS OF INTERVAL SETTING FOR REMOVED LAYERS

In the experiments, we set $k = 4$ as the interval for the removed layers. To test the impact of different values of $k$ on our proposed Replacement Learning, we conduct multiple comparative experiments on the CIFAR-10 Krizhevsky et al. (2009) dataset using ViT-Tiny/8 Dosovitskiy et al. (2021) and ResNet-110 He et al. (2016).

As observed in Table 5, when $k = 2$, a larger number of layers are removed, resulting in greater GPU memory savings and a significant reduction in training time. However, this also leads to a reduction in the amount of learned information, which negatively impacts accuracy. When $k = 6$, although the network performs well in terms of performance, it falls short in resource savings. Through comparison, we find that $k = 4$ strikes the best balance between accuracy and resource efficiency.

### 3.3.3 COMPARISON OF FEATURES IN DIFFERENT METHODS

To showcase the advanced capabilities of Replacement Learning, we conduct feature map analyses on CIFAR-10 Krizhevsky et al. (2009) with ResNet-32 He et al. (2016). The resulting figures can be found in Figure 3.

Table 5: Performance comparison on CIFAR-10 with different $k$ setting.

| Backbone | $k$ value setting | Test Accuracy (%) | GPU Memory (GB) | Training Time (sec) |
|---|---|---|---|---|
| ResNet-110 | k=2 | 81.58±1.89 | 6.25 | 18.05 |
|  | k=4 | 83.95±1.17 | 7.62 | 20.93 |
|  | k=6 | 84.08±1.04 | 8.63 | 23.94 |
| ViT-Tiny/8 | k=2 | 71.48±2.39 | 1.70 | 5.19 |
|  | k=4 | 73.71±1.08 | 2.08 | 5.65 |
|  | k=6 | 73.94±1.17 | 2.39 | 6.39 |

Upon analyzing them, we can observe that (a) and (c), which use End-to-End training, are concentrated in specific regions, indicating the presence of significant information within those areas. Conversely, after using Replacement Learning, (b) and (d) capture more comprehensive global features, including localized edge features. It follows that our method can compensate for the shortcomings of other methods.

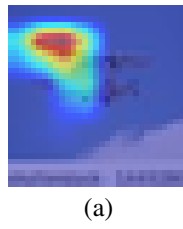 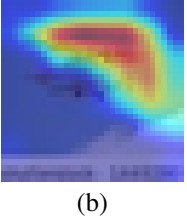 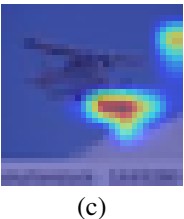 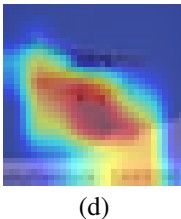

(a)  (b)  (c)  (d)

Figure 3: Visualization of feature maps. (a) Feature map of ResNet-32 with End-to-End training. (b) Feature map of ResNet-32 with Replacement Learning. (c) Feature map of ViT-Tiny/8 with End-to-End training. (d) Feature map of ViT-Tiny/8 with Replacement Learning.

### 3.3.4 COMPARISON OF USING DIFFERENT PARTS OF PARAMETERS

To further validate the importance of leveraging the parameters from preceding and succeeding layers, we conducted an ablation study. Following the main experimental setup, we used ViT-T/8 as the backbone on the CIFAR-10 dataset. Specifically, we compared the results under four configurations: (i) using both attention parameters (including the qkv and $W_o$ layers) and MLP parameters, (ii) using only attention parameters, (iii) using only MLP parameters, and (iv) not using any parameters from adjacent layers. The results in Table 6 indicate that incorporating more parameters consistently leads to better performance. Moreover, attention parameters contribute more significantly than MLP parameters, while excluding all parameters causes a substantial performance drop.

Table 6: Ablation of Parameters in Computing Layers.

| Method | Accuracy | GPU Memory | Training Time |
|---|---|---|---|
| RepL | 73.71±1.08 | 2.08G | 5.65s |
| RepL (only Attention weights) | 72.39±0.97 | 2.05G | 5.59s |
| RepL (only MLP weights) | 72.14±1.34 | 2.07G | 5.53s |
| RepL (no weights) | 69.30±2.11 | 2.02G | 5.20s |

### 3.3.5 COMPARISON OF USING DIFFERENT LAYERS

To validate our design, we conduct experiments with ResNet-110 He et al. (2016) and ViT-Tiny/8 Dosovitskiy et al. (2021) as the backbones, using End-to-End training Rumelhart et al. (1985) as the baseline, and comparing three methods for the computing layers: outputs from the preceding layer, outputs from the succeeding layer, and outputs from both the preceding and succeeding layers.

As shown in Table 7, when using only the outputs from either the previous or the subsequent layer, there is a noticeable decline in accuracy. In contrast, utilizing both the preceding and succeeding

layers simultaneously enhances the model's performance, surpassing that of traditional End-to-End training Rumelhart et al. (1985). This demonstrates the importance of balancing historical and new information in the design of Replacement Learning, which has a positive impact on model performance.

Table 7: Performance comparison on CIFAR-10 using different layers.

| ResNet-110 | | | ViT-Tiny/8 | | |
|---|---|---|---|---|---|
| Preceding Layer | Succeeding Layer | Test Accuracy (%) | Preceding Layer | Succeeding Layer | Test Accuracy (%) |
| ✗ | ✗ | 83.21±1.29 | ✗ | ✗ | 72.77±1.31 |
| ✗ | ✓ | 82.14±2.38 | ✗ | ✓ | 72.18±1.93 |
| ✓ | ✗ | 79.56±3.31 | ✓ | ✗ | 69.37±4.85 |
| ✓ | ✓ | **83.95±1.17** | ✓ | ✓ | **73.71±1.08** |

### 3.4 DETECTION EXPERIMENTS AND ANALYSIS

To evaluate the performance of Replacement Learning on other tasks, we conduct experiments on the COCO dataset Lin et al. (2015) using RetinaNet-R50 and RetinaNet-R101 Lin et al. (2018) as backbones. In these experiments, we utilize 4 Nvidia A100 GPUs, with a batch size of 8, a learning rate of 4e-5, and the Adam optimizer. The training is carried out for a total of 100 epochs. Detailed results can be found in Table 8.

Table 8: Performance comparison on COCO using different backbones. * means the addition of Replacement Learning.

| Backbone | mAP | AP@50 | AP@75 | GPU Memory (GB) | Training Time (sec) |
|---|---|---|---|---|---|
| RetinaNet-R50 | 30.42 | 51.72 | 30.80 | 6.85 | 3859.11 |
| RetinaNet-R50* | 30.64(↑0.22) | 52.44(↑0.72) | 31.15(↑0.35) | 5.82(↓15.04%) | 3245.23(↓15.91%) |
| RetinaNet-R101 | 32.36 | 54.21 | 32.91 | 8.19 | 5548.09 |
| RetinaNet-R101* | 32.76(↑0.40) | 54.80(↑0.59) | 32.98(↑0.07) | 6.65(↓18.80%) | 4671.33(↓15.80%) |

The table illustrates that the Replacement Learning model demonstrates significant performance improvements across various depth detection models, while concurrently reducing both GPU memory usage and training time. These results underscore the effectiveness and efficiency of the proposed method, confirming its versatility in addressing a broad spectrum of deep learning tasks with diverse requirements.

## 4 CONCLUSION

This paper introduces a novel learning approach called Replacement Learning, designed to address the challenge of maintaining model performance while reducing computational overhead and resource consumption. Replacement Learning effectively reduces the parameter count by removing specific layers and replacing them with computing layers. These computing layers integrate the outputs of the preceding and subsequent layers, enhancing the integration of low-level and high-level features, thereby improving the overall performance of the model. We apply Replacement Learning to various model architectures with different depths and evaluate their performance on five widely used datasets in classification and object detection tasks. The results demonstrate that the proposed Replacement Learning not only reduces training time and GPU usage but also consistently outperforms end-to-end training in terms of overall performance.

**Limitations and future work:** While Replacement Learning reduces parameter computation, saves memory, and shortens training time, all while outperforming End-to-End training, it has only been tested on image-based tasks. It has yet to be applied to larger models in natural language processing or multimodal settings. Future work will explore the impact of Replacement Learning on these tasks to provide a more comprehensive evaluation of its effectiveness.

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

## A    APPENDIX

### A.1    USE OF LLMS

In the appendix's theoretical analysis section, to verify the mathematical soundness and symbolic accuracy of a few selected formulas.

### A.2    RELATED WORK

**Alternatives to backpropagation.** To address the limitations of backpropagation, such as high computational cost, various alternative methods have been proposed, including target propagation Lee et al. (2015); Bartunov et al. (2018), feedback alignment Lillicrap et al. (2014); Nøkland (2016), and decoupled neural interfaces (DNI) Jaderberg et al. (2017). These approaches bypass traditional global backpropagation by directly propagating errors to individual layers, reducing memory usage and enhancing efficiency. Forward gradient learning Dellaferrera & Kreiman (2022); Ren et al. (2022) offers a new paradigm for training deep networks more effectively. Local learning Zhang et al. (2024); Zhu et al. (2024) segments the network into smaller, independently trained modules, optimizing local objectives to lower computational demands while preserving some global features Su et al. (2024a;b). However, excessive segmentation can lead to coordination issues, harming overall performance, especially on complex datasets like ImageNet.

**Utilizing surrounding layers.** Leveraging the high similarity in learning conditions of surrounding layers, researchers have solved many problems in deep learning. Some studies have applied Residual Networks (ResNets) He et al. (2016), by adding a shortcut connection to the activation function of the next layer, this identity mapping enables ResNet to address the issues of degradation Philipp et al. (2018); Borawar & Kaur (2023), enhancing both the convergence speed and accuracy of the network Zhang et al. (2019); Allen-Zhu & Li (2019). Additionally, some researchers have proposed skipping attention, reusing the self-attention calculations from one layer in the approximations for attention in subsequent layers, achieving higher throughput Venkataramanan et al. (2023). However, due to the repeated use of prior layers, this method carries the risk of error propagation and could potentially cause losses during the learning process, impacting the model's generalization ability.

### A.3    EXPERIMENTAL SETUP DETAILS

We conducted experiments on small-scale datasets (CIFAR-10 Krizhevsky et al. (2009), SVHN Netzer et al. (2011), and STL-10 Coates et al. (2011)) using ViT-Tiny/8 Dosovitskiy et al. (2021), ResNet-32, and ResNet-110 He et al. (2016), with training performed on a single Nvidia A100 GPU. For the ViT models, we used a batch size of 512, the AdamW optimizer, and set the learning rate to 1e-3, training for 250 epochs. For the ResNet models, the batch size was set to 1024, using the SGD optimizer with a learning rate of 0.8, trained for 250 epochs. We follow these augmentation strategies: CIFAR-10: 4-pixel reflection padding followed by random cropping back to 32×32, and horizontal flipping with a probability of 0.5; SVHN: random cropping to 32×32 (with 2-pixel padding), without horizontal flipping; STL-10: random cropping to 96×96 (with 4-pixel padding) and horizontal flipping with a probability of 0.5. On the ImageNet dataset Deng et al. (2009), we conducted experiments using 4 Nvidia A100 GPUs for ViT-Tiny/16 and ViT-Small/16 Dosovitskiy et al. (2021), with a batch size of 1024, the AdamW optimizer, and a learning rate of 7.5e-4. For the ResNet models (ResNet-34, ResNet-101, and ResNet-152 He et al. (2016)), we used a batch size of 512, the SGD optimizer, and set the learning rate to 0.2, training for 90 epochs. For training samples, we use a $224 \times 224$ random crop with random horizontal flips, while for test samples, we apply a $224 \times 224$ resize followed by a central crop.

### A.4    COMPARISON OF THE DISTRIBUTION OF CLASSIFIED DATA POINTS

To compare E2E Training Rumelhart et al. (1985) and Replacement Learning in feature learning, we perform t-SNE visualization Van der Maaten & Hinton (2008) on ResNet-110 He et al. (2016) using the SVHN dataset Netzer et al. (2011), as shown in Figure 4. In the t-SNE plot for End-to-End training (a), significant overlap between target and non-target classes indicates poor class discrimination. In contrast, the Replacement Learning visualization (b) shows more compact and distinct target class clusters, with clearer boundaries between target and non-target classes, reducing inter-class confusion.

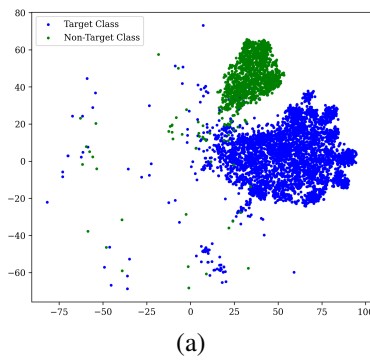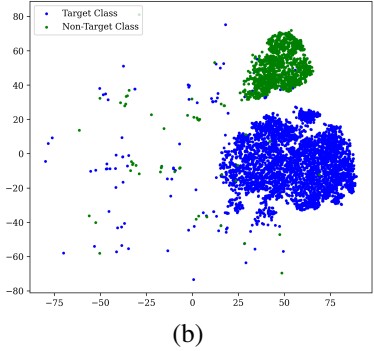

(a)                                                        (b)

Figure 4: T-SNE visualization. (a) is t-SNE of E2E training, (b) is t-SNE of Replacement Learning.

These results demonstrate the superior classification performance of Replacement Learning over End-to-End training Rumelhart et al. (1985).

### A.5 SUPPLEMENTARY EXPERIMENTS

#### A.5.1 COMPARATIVE EXPERIMENTS WITH RELATED METHODS

To verify the generality of our approach, we compared it against Stochastic Depth Huang et al. (2016) and Checkpointing Chen et al. (2016), and further combined our method with these two techniques. The experimental results are illustrated in the following Table. 9.

Table 9: Comparative Experiments with Stochastic Depth and Checkpointing, the results in the table are based on a single run.

| Dataset | Backbone | Method | Acc@1 | GPU (GB) | Time (s/epoch) |
|---------|----------|--------|-------|----------|----------------|
| CIFAR-10 | ResNet-32 | E2E | 93.25 | 3.38 | 5.24 |
| | | RepL | 93.29 | 2.69 | 4.37 |
| | | Stochastic Depth | 93.04 | 3.31 | 5.05 |
| | | RepL+Stochastic Depth | 93.17 | 2.67 | 4.18 |
| | | Checkpointing | 93.13 | 1.77 | 8.74 |
| | | RepL+Checkpointing | 93.24 | 1.64 | 7.22 |
| ImageNet | ResNet-101 | E2E | 78.19 | 20.95 | 720 |
| | | RepL | 78.43 | 18.01 | 616 |
| | | Stochastic Depth | 77.63 | 19.39 | 652 |
| | | RepL+Stochastic Depth | 78.11 | 17.12 | 551 |
| | | Checkpointing | 78.25 | 14.47 | 1012 |
| | | RepL+Checkpointing | 78.29 | 12.93 | 819 |

#### A.5.2 EXPERIMENTS ON THE NLP TASK

We conduct the experiments on the NLP model, and the experimental configuration and results are shown in the table 10 below. The tokenization method adopts basic English tokenization. In the process of building the vocabulary, only words with an occurrence frequency of no less than 2 are retained. Meanwhile, the <eos> token is appended at the end of each sentence. For sequence segmentation, the backpropagation through time with a length of 128 is used. The experiment was trained for 20 epochs, and the significant variance was obtained through 5 experiments (different seeds).

Table 10: Performance on WikiText-2 using Transformer-LM-12L-512d-8H-2048ff.

| Dataset | Model | Method | Test PPL ($\downarrow$) | GPU Memory (GB) | Time (per epoch, sec) |
|---|---|---|---|---|---|
| WikiText-2 | Transformer -LM-12L-512d-8H-2048ff | E2E | 195.42±1.84 | 10.92 | 20.8 |
| | | RepL | 193.31±3.39 | 9.61 | 17.7 |
| | Configuration | Hardware: Single A100 Batch size: 64 Optimizer: AdamW Learning rate: 3e-4 | Grad_clip: 1.0 Weight decay: 0.01 fp: 16 | | |

### A.5.3 INFERENCE ON IMAGENET

We have conducted experiments on inference throughput, and the results are presented in the Table. 11. We used a single GPU, and the batch size is 128.

Table 11: Results on the GPU Memory Usage and Time during inference on ResNet-101 and ViT-S/16.

| Dataset | Backbone | Method | GPU Memory | Time |
|---|---|---|---|---|
| ImageNet | ResNet-101 | E2E | 3.97G | 39.12s |
| | | RepL | 3.65G | 36.26s |
| | ViT-S/16 | E2E | 2.69G | 48.29s |
| | | RepL | 2.45G | 41.42s |

### A.5.4 FINE-TUNING ON VITS

To verify the effectiveness of RepL in the finetuning setting, we conduct experiments on CIFAR-10, SVHN, and STL-10 using pretrained weights obtained from ImageNet-1K. The experimental settings were: batch size = 512, learning rate = 2e-4, optimizer = AdamW, and epochs = 100. The results are summarized in Table 12.

Table 12: Finetune results on ViT-S/16.

| Datasets | Model | Method | Acc@1 | GPU Memory (GB) | Time (per epoch) |
|---|---|---|---|---|---|
| CIFAR-10 | ViT-S/16 | E2E | 95.66 | 25.56 | 32.45 |
| | | RepL | 95.89 | 20.14 | 25.18 |
| SVHN | ViT-S/16 | E2E | 96.92 | 25.56 | 48.44 |
| | | RepL | 96.97 | 20.14 | 38.01 |
| STL-10 | ViT-S/16 | E2E | 94.88 | 25.56 | 5.91 |
| | | RepL | 95.11 | 20.14 | 4.66 |

### A.5.5 FINE-TUNING FOR DOWNSTREAM TASKS

We fine-tuned the pre-trained model (ImageNet-1k Deng et al. (2009), trained with RepL) on the CityScapes dataset using the SGD optimizer with a batch size of 16, a learning rate of 0.1, a crop size of 768, and trained for 30k iterations (about 164 epochs) on a single GPU. The experimental results are shown in the following table 13.

When fine-tuning for downstream tasks, RepL does not compromise transfer learning performance. First, its computational layers preserve the core feature patterns acquired by the model through parameter fusion of adjacent layers, rather than randomly pruning information. Second, parameter reduction mitigates overfitting risks during fine-tuning, particularly evident in low-data scenarios. Finally,

learnable blocks dynamically adjust the weight contributions between preceding and succeeding layers during fine-tuning, enhancing task-specific feature representation.

Table 13: Performance comparison on CityScapes using different backbones.

| Backbone | Method | Overall Accuracy | Mean Accuracy | Mean IoU | GPU Memory (GB) | Time (per epoch, sec) |
|---|---|---|---|---|---|---|
| DeepLabV3-R50 | E2E | 95.27 | 80.83 | 73.34 | 23.90 | 80 |
| | RepL | 95.32 | 81.14 | 73.81 | 20.28 | 68 |
| DeepLabV3Plus-R50 | E2E | 95.66 | 81.89 | 74.61 | 26.81 | 82 |
| | RepL | 95.71 | 82.21 | 75.25 | 22.67 | 69 |
| DeepLabV3-R101 | E2E | 95.51 | 82.31 | 74.41 | 30.91 | 95 |
| | RepL | 95.54 | 82.71 | 74.55 | 25.90 | 82 |
| DeepLabV3Plus-R101 | E2E | 95.84 | 83.24 | 75.53 | 34.42 | 101 |
| | RepL | 95.89 | 84.02 | 76.31 | 28.92 | 86 |

### A.5.6 EXTRA ABLATION STUDY ON ViT

In our ViT experiments, RepL employs two learnable parameters, $\alpha$ and $\beta$, to fuse the parameters from the preceding and succeeding layers, respectively. To validate that using two learnable parameters is indeed more effective than a single one, we conducted an ablation study. As shown in Table. 14, introducing both $\alpha$ and $\beta$ does not incur any additional GPU memory consumption or training time. Moreover, this configuration consistently achieves noticeably better performance compared to using a single learnable parameter.

Table 14: Ablation on number of parameters in RepL. We use ViT-T/8 on CIFAR-10 dataset.

| Method | Accuracy | GPU Memory | Training Time |
|---|---|---|---|
| RepL(2 parameter) | $73.71 \pm 1.08$ | 2.08G | 5.65s |
| RepL(1 parameter) | $73.09 \pm 0.85$ | 2.08G | 5.65s |

### A.6 PARAMETER ANALYSIS

We quantify how many learnable weights are discarded by Replacement Learning and how many new ones are introduced. Let a network contain $n$ layers, indexed from 1 to $n$. Denote by $P_i := \|W_i\|_0$ the number of parameters of the $i$-th layer,[2] and let $P_{\text{tot}}^{\text{E2E}} := \sum_{i=1}^{n} P_i$ be the parameter count of ordinary end-to-end training.

**Replacement Learning with removal interval $k$.** A fraction $\gamma := |\mathcal{F}|/n = \lfloor \frac{n}{k} \rfloor / n \approx \frac{1}{k}$ of the layers are removed. The *retained* parameters are therefore $(1 - \gamma) P_{\text{tot}}^{\text{E2E}}$.

**CNNs.** For every removed layer $i \in \mathcal{F}$ two depth-wise $1 \times 1$ convolutions are inserted, contributing

$$\underbrace{C_{i-1}^{\text{out}}}_{\phi_{i-1}} + \underbrace{C_{i+1}^{\text{out}}}_{\phi_{i+1}} \quad \text{weights.} \tag{13}$$

*Upper bound.* Because $C_{i\pm1}^{\text{out}} \le \max_j C_j^{\text{out}}$, the total number of *new* weights satisfies

$$P_{\text{add}}^{\text{CNN}} \le 2\gamma n \max_j C_j^{\text{out}} = \frac{2n}{k} C_{\max}. \tag{14}$$

Since a normal $k \times k$ convolution carries $C_i^{\text{out}} C_i^{\text{in}} k^2$ parameters, one obtains the *global* bound

---

[2]For CNNs $P_i = C_i^{\text{out}} C_i^{\text{in}} k^2$; for ViTs it is the sum of the projection matrices of the $i$-th transformer block.

$$P_{\text{tot}}^{\text{RepL}} \leq (1 - \gamma) P_{\text{tot}}^{\text{E2E}} + \tfrac{2n}{k} C_{\max} < \left(1 - \tfrac{1}{k}\right) P_{\text{tot}}^{\text{E2E}} + \mathcal{O}(nC_{\max}). \tag{15}$$

**ViTs.** Each removed transformer block contributes exactly two learnable parameters, hence

$$P_{\text{add}}^{\text{ViT}} = 2\gamma n = \tfrac{2n}{k}, \qquad P_{\text{tot}}^{\text{RepL}} = (1 - \gamma) P_{\text{tot}}^{\text{E2E}} + \tfrac{2n}{k}. \tag{16}$$

**Tightness.** If all $P_i$ are identical ($P_i \equiv \bar{P}$) one has $P_{\text{tot}}^{\text{E2E}} = n\bar{P}$ and $P_{\text{tot}}^{\text{RepL}} = (1 - \gamma)n\bar{P} + P_{\text{add}}$, so the relative reduction is bounded by

$$\frac{P_{\text{tot}}^{\text{RepL}}}{P_{\text{tot}}^{\text{E2E}}} = 1 - \frac{1}{k} + \mathcal{O}\left(\frac{1}{n}\right) \qquad \text{(CNN \& ViT)}. \tag{17}$$

Thus, Replacement Learning discards *at least* $1/k$ of the original parameters and its overhead decays as $n$ grows.

## A.7 COMPLEXITY ANALYSIS

We analyse the change in *floating-point operations* (FLOPs) and *activation memory* during one training iteration.

### A.7.1 FLOPs

**CNNs.** A standard $k \times k$ convolution with stride 1 on a feature map of size $H \times W$ costs

$$F_{\text{conv}} = 2 C^{\text{in}} C^{\text{out}} k^2 HW. \tag{18}$$

At a replaced site, the learnable blocks $\mathcal{T}_{i-1}, \mathcal{T}_{i+1}$ act in *weight space* and introduce no per-pixel cost. At run time we apply a single $1 \times 1$ convolution $\widehat{W}_i \in \mathbb{R}^{C_{i+1}^{\text{in}} \times C_{i-1}^{\text{in}} \times 1 \times 1}$:

$$F_{\text{RepL}}^{\text{CNN}} = 2 C_{i-1}^{\text{in}} C_{i+1}^{\text{in}} HW. \tag{19}$$

Since $k > 1$ and typically $C_{i\pm1}^{\text{in}} \approx C_{i\pm1}^{\text{out}}$,

$$\frac{F_{\text{RepL}}^{\text{CNN}}}{F_{\text{conv}}} = \frac{C_{i-1}^{\text{in}} C_{i+1}^{\text{in}}}{C_i^{\text{in}} C_i^{\text{out}} k^2} \leq \frac{1}{k^2}. \tag{20}$$

Replacing a fraction $\gamma \approx \tfrac{1}{k}$ of blocks yields the network-level bound

$$F_{\text{tot}}^{\text{RepL}} \leq (1 - \gamma) F_{\text{tot}}^{\text{E2E}} + \gamma \tfrac{1}{k^2} F_{\text{tot}}^{\text{E2E}} = \left(1 - \tfrac{1}{k} + \tfrac{1}{k^3}\right) F_{\text{tot}}^{\text{E2E}}. \tag{21}$$

**ViTs.** Let a standard transformer block cost $F_{\text{SA}}$ FLOPs (self-attention + MLP). At a replaced site, the learnable block is implemented by two scalars $(\alpha_i, \beta_i)$ and executes only two $d \times d$ linear maps on all $T$ tokens:

$$F_{\text{RepL}}^{\text{ViT}} = 2 \cdot (2d^2 T) = 4d^2 T, \tag{22}$$

thus

$$F_{\text{tot}}^{\text{RepL}} \leq (1 - \gamma) F_{\text{tot}}^{\text{E2E}} + \gamma \cdot \frac{4d^2 T}{F_{\text{SA}}} F_{\text{tot}}^{\text{E2E}} < \left(1 - \tfrac{1}{k}\right) F_{\text{tot}}^{\text{E2E}}, \tag{23}$$

because $F_{\text{SA}} \gg 4d^2 T$ in practice.

### A.7.2 ACTIVATION / MEMORY FOOTPRINT

During training, removing a convolutional or transformer block also removes its checkpointed *input* activation for backprop. Let $A_i$ be the size (bytes) of the input activation to block $i$. The E2E peak is $M_{\text{peak}}^{\text{E2E}} = \max_i \sum_{j \leq i} A_j$. RepL discards every $k$-th block from the executed path; the learnable blocks act in weight space and add no extra feature maps. Hence

$$M_{\text{peak}}^{\text{RepL}} \leq \left(1 - \tfrac{1}{k}\right) M_{\text{peak}}^{\text{E2E}} + \mathcal{O}\left(\tfrac{n}{k}\right) \cdot \underbrace{\text{(LN/BN stats)}}_{\text{negligible}}, \tag{24}$$

which is consistent with the empirical $15\%-26\%$ GPU-memory reduction.

**Discussion.** Eq. (15)–(23) show that, for both CNNs and ViTs, Replacement Learning enjoys *linear* savings in parameters, FLOPs and peak memory with respect to the removal rate $\frac{1}{k}$, while introducing only $\mathcal{O}(\frac{n}{k})$ extra learnable parameters or depth-wise kernels. These tight bounds theoretically explain the consistent empirical gains observed across all datasets and model families.

## A.8   ERROR BOUND & CONVERGENCE ANALYSIS

**Additional notation.** Let $F(\mathbf{x};\theta) = f_n \circ \cdots \circ f_1(\mathbf{x})$ be the *baseline* network and $\widehat{F}(\mathbf{x};\theta,\psi)$ its *Replacement Learning* variant, where $\psi$ collects all learnable–block parameters. Denote the loss by $\mathcal{L}(\cdot,y)\colon \mathbb{R}^{d_o}\to\mathbb{R}$, and write $\ell(\theta):=\mathbb{E}_{(\mathbf{x},y)}\,\mathcal{L}\big(F(\mathbf{x};\theta),y\big)$ and $\widehat{\ell}(\theta,\psi):=\mathbb{E}_{(\mathbf{x},y)}\,\mathcal{L}\big(\widehat{F}(\mathbf{x};\theta,\psi),y\big)$.

### A.8.1   APPROXIMATION BIAS OF A COMPUTING LAYER

**Definition 1** (Local operator deviation). Let $g_i(\cdot)$ be the (linear part of the) original block-$i$ map before its normalization/nonlinearity, and $\widehat{g}_i(\cdot)$ be the corresponding map produced by the learnable block (i.e., $\widehat{g}_i(\mathbf{h}) = \widehat{W}_i\mathbf{h}$ for CNNs and $\widehat{g}_i(\mathbf{h}) = \widehat{A}_i\mathbf{h} + \widehat{M}_i\mathbf{h}$ for ViTs). Define the operator-norm deviation

$$\varepsilon_i \;:=\; \sup_{\mathbf{h}\neq 0}\frac{\|\widehat{g}_i(\mathbf{h}) - g_i(\mathbf{h})\|}{\|\mathbf{h}\|}, \qquad \varepsilon_{\max} = \max_{i\in\mathcal{F}}\varepsilon_i. \tag{25}$$

This avoids shape-mismatch issues and subsumes the CNN alignment maps $\mathcal{T}_{i\pm 1}$ implicitly through $\widehat{g}_i$.

**Lemma 1** (Layer-wise output deviation). *If each block (including its normalization/nonlinearity) is $L$-Lipschitz, then for any input $\mathbf{x}$,*

$$\big\|\widehat{F}(\mathbf{x};\theta,\psi) - F(\mathbf{x};\theta)\big\| \;\leq\; L^{|\mathcal{F}|}\,\varepsilon_{\max}\,\max_{i\in\mathcal{F}}\|\mathbf{h}_{i-1}\|. \tag{26}$$

*Proof.* Insert $\widehat{g}_i = g_i + (\widehat{g}_i - g_i)$ into the forward recursion at replaced sites and propagate Lipschitz bounds. □

### A.8.2   GRADIENT BIAS AND STABLE TRAINING

**Lemma 2** (Gradient deviation). *Let every composite function up to layer $j$ be $L$-smooth[3]. Then*

$$\big\|\nabla_\theta\widehat{\ell}(\theta,\psi) \;-\; \nabla_\theta\ell(\theta)\big\| \;\leq\; L\,H_{\max}\,\varepsilon_{\max}. \tag{27}$$

*Proof.* Using Lemma 1 and $L$-smoothness of the composite loss, $\|\nabla\mathcal{L}(\widehat{F}) - \nabla\mathcal{L}(F)\| \leq L\|\widehat{F} - F\|$. Take expectation over the data. □

### A.8.3   CONVERGENCE UNDER SGD AND ADAM

**Setup.** Let $F(\mathbf{x};\theta) = f_n \circ \cdots \circ f_1(\mathbf{x})$ be the baseline network and $\widehat{F}(\mathbf{x};\theta,\psi)$ the variant trained with learnable blocks, where $\psi$ collects all learnable–block parameters. Given a sample $(\mathbf{x},y)$ and a loss $\mathcal{L}(\cdot,y)$, define the population objectives

$$\ell(\theta):=\mathbb{E}_{(\mathbf{x},y)}\big[\mathcal{L}\big(F(\mathbf{x};\theta),y\big)\big], \qquad \widehat{\ell}(\theta,\psi):=\mathbb{E}_{(\mathbf{x},y)}\big[\mathcal{L}\big(\widehat{F}(\mathbf{x};\theta,\psi),y\big)\big].$$

**Assumptions.** We make the following standard conditions used in nonconvex analyses:

(A1) Each $f_j$ is $L$-smooth and $G$-Lipschitz; $\mathcal{L}(\cdot,y)$ is $L$-smooth.

(A2) Mini-batch gradients are unbiased with variance $\sigma^2$: $\mathbb{E}[g_t] = \nabla\widehat{\ell}(\theta_t,\psi_t)$ and $\mathbb{E}\|g_t - \nabla\widehat{\ell}(\theta_t,\psi_t)\|^2 \leq \frac{\sigma^2}{B}$ for batch size $B$.

---

[3]$g$ is $L$-smooth if $\|\nabla g(a) - \nabla g(b)\| \leq L\|a - b\|$.

(A3) (Bounded synthesis bias) For every removed index $i \in \mathcal{F}$, the learnable-block synthesis error on weights is bounded in Frobenius norm by $\varepsilon$; equivalently, the *induced* gradient bias satisfies $\left\| \nabla_\theta \widehat{\ell}(\theta, \psi) - \nabla_\theta \ell(\theta) \right\| \leq c\,\varepsilon$ for some constant $c$ depending on $(L, G)$ (Lemma 3).

**Lemma 3** (Gradient bias induced by learnable blocks). *Assume the forward discrepancy introduced at removed sites is bounded as $\|\widehat{F}(\mathbf{x}; \theta, \psi) - F(\mathbf{x}; \theta)\| \leq H_{\max}\,\varepsilon$ for all $(\mathbf{x}, y)$, where $H_{\max}$ upper-bounds the relevant activations. If $\mathcal{L}(\cdot, y)$ is $L$-smooth, then*

$$\left\| \nabla_\theta \widehat{\ell}(\theta, \psi) - \nabla_\theta \ell(\theta) \right\| \ \leq \ L\,H_{\max}\,\varepsilon.$$

*Proof.* By $L$-smoothness of $\mathcal{L}(\cdot, y)$ and the chain rule, $\left\| \nabla \mathcal{L}(\widehat{F}) - \nabla \mathcal{L}(F) \right\| \leq L \|\widehat{F} - F\|$ pointwise; take expectation and use the assumed forward bound. $\qquad \blacksquare$

**Theorem 2** (SGD convergence with learnable blocks). *Under (A1)–(A3), run SGD on $(\theta, \psi)$ with step sizes $\eta_t = \frac{\eta}{\sqrt{t}}$ for $T$ steps. Let $\ell^\star := \inf_\theta \ell(\theta)$ and denote the constant $\beta := L\,H_{\max}$ from Lemma 3. Then*

$$\frac{1}{T} \sum_{t=1}^{T} \mathbb{E}\left[ \|\nabla \widehat{\ell}(\theta_t, \psi_t)\|^2 \right] \ \leq \ \underbrace{\frac{2(\ell_0 - \ell^\star)}{\eta \sqrt{TB}} + \frac{\eta L \sigma^2}{B}}_{\text{standard nonconvex SGD Ghadimi \& Lan (2013)}} \ + \ \underbrace{2\,\beta\,\varepsilon}_{\text{bias from learnable blocks}} \quad . \quad (28)$$

*Sketch.* Follow the descent-lemma proof for nonconvex SGD Ghadimi & Lan (2013) but write the update in terms of the *perturbed* gradient $\nabla \widehat{\ell} = \nabla \ell + b$, where $\|b\| \leq \beta\,\varepsilon$ by Lemma 3. The cross term contributes an additive constant $\mathcal{O}(\beta\varepsilon)$ that telescopes to $2\beta\varepsilon$ in the averaged bound, yielding equation 28. $\qquad \blacksquare$

**Corollary 1** (Adam/AdamW). *If Adam is used with AMSGrad-style conditions ensuring convergence in the nonconvex setting (e.g., Reddi et al. (2018)), or AdamW with standard assumptions Tran-Dinh et al. (2021), then the iterates satisfy*

$$\min_{1 \leq t \leq T} \ \mathbb{E}\left[ \|\nabla \widehat{\ell}(\theta_t, \psi_t)\|^2 \right] \ = \ \widetilde{\mathcal{O}}\big(T^{-\frac{1}{2}}\big) \ + \ \mathcal{O}(\varepsilon),$$

*i.e., the usual $T^{-\frac{1}{2}}$ decay up to an additive term that is linear in the bounded synthesis bias $\varepsilon$.*

**Remarks.** (i) When the learnable blocks synthesize weights with vanishing error ($\varepsilon \to 0$), the bounds reduce to the classical rates. (ii) For fixed replacement interval $k$ and stable training, $\varepsilon$ is a small constant determined by how well neighbor-conditioned synthesis approximates the removed operator; the asymptotic $T^{-\frac{1}{2}}$ behavior is therefore preserved while enjoying lower per-epoch cost. (iii) The bounds are agnostic to the CNN/ViT instantiation; only the magnitude of $\varepsilon$ changes with the specific synthesis rule (Sec. 2.2-2.3).

## A.9 MULTI-REPLACEMENT ERROR PROPAGATION

Let $\mathcal{F} \subset \{1, \ldots, n\}$ be the set of replaced indices and assume each full block (including normalization and nonlinearity) is $L$-Lipschitz. For $i \in \mathcal{F}$ let $\varepsilon_i$ be the local operator deviation defined in Eq. equation 25. Denote by $r := |\mathcal{F}|$ and $\bar{\varepsilon} := \frac{1}{r} \sum_{i \in \mathcal{F}} \varepsilon_i$, $\varepsilon_{\max} := \max_{i \in \mathcal{F}} \varepsilon_i$.

**Proposition 3** (Accumulated output deviation). *For any input $\mathbf{x}$,*

$$\left\| \widehat{F}(\mathbf{x}; \theta, \psi) - F(\mathbf{x}; \theta) \right\| \ \leq \ \begin{cases} L^r\,\varepsilon_{\max} \max_{i \in \mathcal{F}} \|\mathbf{h}_{i-1}\|, & \text{(worst-case bound)} \\ \dfrac{1 - L^r}{1 - L}\,\bar{\varepsilon} \max_{i \in \mathcal{F}} \|\mathbf{h}_{i-1}\|, & \text{if } L < 1. \end{cases}$$

*Proof.* Insert $\widehat{g}_i = g_i + (\widehat{g}_i - g_i)$ at each $i \in \mathcal{F}$ and propagate perturbations. For $L < 1$ the series of perturbations forms a geometric sum. $\qquad \blacksquare$

**Implication.** When blocks are *non-expansive* ($L \leq 1$), e.g., with post-normalization, the accumulated discrepancy grows at most linearly with $r$ and is further damped if $L < 1$. This complements Lemma 1 by accounting for multiple replacements.

## A.10 RECOVERABILITY AND EXPRESSIVITY OF THE COMPUTING LAYER

We formalize when the learnable block can *exactly* reproduce the removed operator ($\varepsilon_i = 0$), and what subspace of operators it can represent.

**CNN case.** Let the linear part of the removed site be a map $g_i(\mathbf{h}) = W_i\mathbf{h}$ (after any fixed alignment used by the baseline). The learnable block synthesizes $\widehat{W}_i = \mathcal{T}_{i-1}(W_{i-1}) + \mathcal{T}_{i+1}(W_{i+1})$, where $\mathcal{T}_{i\pm1}$ act on the channel modes of their kernel tensors. Define the *synthesis span*

$$\mathcal{S}_i := \left\{ \mathcal{T}_{i-1}(U) + \mathcal{T}_{i+1}(V) \ : \ U \in \mathcal{U}_{i-1}, \ V \in \mathcal{U}_{i+1} \right\},$$

where $\mathcal{U}_{i\pm1}$ denote the admissible weight tensors with the same shape as $W_{i\pm1}$.

**Lemma 4** (Exact recoverability in CNNs). *If $W_i \in \mathcal{S}_i$, then there exist learnable-block parameters such that $\widehat{W}_i = W_i$ and thus $\varepsilon_i = 0$.*

*Proof.* By definition of $\mathcal{S}_i$ there exist $U^\star, V^\star$ with $W_i = \mathcal{T}_{i-1}(U^\star) + \mathcal{T}_{i+1}(V^\star)$; setting the learnable-block weights to realize $(U^\star, V^\star)$ gives the claim. □

**Rank and span.** Write the $1\times1$ equivalent of the synthesized operator as a matrix $\widehat{W}_i \in \mathbb{R}^{C_{i+1}^{\text{in}} \times C_{i-1}^{\text{in}}}$. Then $\text{rank}(\widehat{W}_i) \leq \text{rank}(\mathcal{T}_{i-1}(W_{i-1})) + \text{rank}(\mathcal{T}_{i+1}(W_{i+1}))$. In typical same-width stages, both terms are full row/column rank, so $\widehat{W}_i$ can achieve full rank and does not bottleneck the channel dimension.

**ViT case.** In ViT instantiation, the replacement operator is explicitly constrained to a 2D neighbor span: $A_i^{\text{RepL}} = \alpha_i A_{i-1} + \beta_i A_{i+1}$, $M_i^{\text{RepL}} = \alpha_i M_{i-1} + \beta_i M_{i+1}$, where $A_{i-1}, A_{i+1} \in \mathbb{R}^{d \times d}$ (resp .$M_{i-1}, M_{i+1}$) are the attention (resp. MLP) operators of the neighboring blocks, and $\alpha_i, \beta_i$ are learned scalars. Thus, by construction, each replaced block is synthesized inside the span of its two neighbors; RepL never introduces an arbitrary new block.

To quantitatively assess how well the original block is captured by this 2D span, we conducted a feature-space span diagnostic on the same ViT-tiny / CIFAR-10 setting used in our main experiments:

**Backbone architecture:** A 12-block ViT-tiny with patch size $8 \times 8$, embedding dimension 192, and 3 heads (the same configuration as in our CIFAR-10 experiments).

**Training setup:** We trained a standard backbone ViT ("bp") with direct backpropagation on CIFAR-10 using the script described in the paper.

**Where we probe:** We focus on the block indices that RepL would remove under the same periodic schedule used in the method (remove every 4th block, excluding the last one). In a 12-block ViT-tiny, this yields removed indices $i = 2, 6, 10(0 - based)$.

**Diagnostic metric:** Let $h_k(x)$ denote the hidden representation after block $k$ for an input $x$. For each removed index $i$, we seek the best approximation $h_i(x) \approx \tilde{\alpha}_i h_{i-1}(x) + \tilde{\beta}_i h_{i+1}(x)$, by solving a least-squares problem over CIFAR-10 samples. From this we compute: the relative reconstruction error $r_i = \frac{\left\| h_i(x) - \Pi_{\text{span}(h_{i-1}(x), h_{i+1}(x))}(h_1(x)) \right\|_2}{\|h_i(x)\|_2}$ aggregated over the dataset. The principal angle (in the 1D case) between $h_i(x)$ and its best neighbor-span reconstruction.

We use CIFAR-10 test images, with the same normalization as training, and run the diagnostic on 20 batches (batch size 512). The results for the two removed blocks are:

**index $i = 2$:**

relative feature-space reconstruction error $r_2 = \mathbf{0.1796}$;

principal angle $= \mathbf{7.63°}$;

fitted coefficients $(\tilde{\alpha}_2, \tilde{\beta}_2) \approx (0.4811, 0.5230)$.

**index** $i = 6$**:**

relative feature-space reconstruction error $r_6 = \mathbf{0.1501}$;

principal angle $= \mathbf{5.22°}$;

fitted coefficients $(\tilde{\alpha}_6, \tilde{\beta}_6) \approx (0.4711, 0.5325)$.

**index i = 10**:

relative feature-space reconstruction error $r_6 = \mathbf{0.1379}$;

principal angle $= \mathbf{4.62°}$;

fitted coefficients $(\tilde{\alpha}_{10}, \tilde{\beta}_{10}) \approx (0.4547, 0.5493)$.

These results show that, measured in terms of their action on real data (hidden representations), the blocks targeted by RepL are well captured by the 2D span of their neighbors: both the relative reconstruction error and the principal angle are small, and the fitted coefficients are close to a symmetric combination of the two neighbors. This empirically supports our modeling choice that a lightweight operator constrained to $\text{span}(A_{i-1}, A_{i+1})$ and $\text{span}(M_{i-1}, M_{i+1})$ is sufficient to replace the original block in ViT backbones.

**Lemma 5** (Exact recoverability in ViTs). *If the linear parts of the removed block satisfy* $A_i \in \text{span}\{A_{i-1}, A_{i+1}\}$ *and* $M_i \in \text{span}\{M_{i-1}, M_{i+1}\}$, *then there exist* $(\alpha_i, \beta_i)$ *such that* $\widehat{A}_i = A_i$ *and* $\widehat{M}_i = M_i$, *so* $\varepsilon_i = 0$.

These statements clarify that $\varepsilon_i$ measures the distance of the removed operator to the neighbor-conditioned synthesis span; when that distance is small (as empirically observed), the induced bias in Sec. A.8 remains negligible.

### A.11    A Simple Compute Accuracy Trade-off for Choosing $k$

Let $C_{\text{epoch}}(k)$ denote the per-epoch training cost (FLOPs or wall time) under interval $k$, and let $\Delta_{\text{acc}}(k)$ denote the excess risk (or a proxy) induced by replacement. From Sec. A.7 we have the approximation $C_{\text{epoch}}(k) \approx \left(1 - \frac{1}{k}\right) C_0$ for a baseline cost $C_0$. From Sec. A.8, the gradient-norm bound adds an $\mathcal{O}(\varepsilon(k))$ bias term. For small replacement rates we model $\varepsilon(k) \approx \frac{c}{k}$ with problem-dependent $c > 0$.

Consider minimizing a weighted objective

$$J(k) \;=\; \lambda\, C_{\text{epoch}}(k) \;+\; \Delta_{\text{acc}}(k), \qquad \Delta_{\text{acc}}(k) \;\approx\; \kappa\, \varepsilon(k) \;=\; \frac{\kappa c}{k},$$

where $\lambda, \kappa > 0$ encode the user's compute/accuracy preference. Using $C_{\text{epoch}}(k) \approx (1 - \frac{1}{k}) C_0$ gives

$$J(k) \;\approx\; \lambda C_0 \left(1 - \frac{1}{k}\right) + \frac{\kappa c}{k} \;=\; \lambda C_0 \;+\; \frac{\kappa c - \lambda C_0}{k}.$$

The surrogate suggests a *threshold* behavior: when $\kappa c < \lambda C_0$, larger $k$ (more aggressive replacement) is favored; otherwise, a smaller $k$ is preferred. In practice, $\kappa c$ can be estimated on a held-out split by measuring the validation loss gap as a function of $k$ for a few short runs, after which $k$ is chosen to meet a compute budget while keeping the additional bias under the tolerance implied by Corollary 1.

### A.12    Bias in practice: empirical $\varepsilon$ via forward and gradient deviations

Our analysis in Section 4 assumes that $\left\|\nabla_\theta \ell_b - \nabla_\theta \ell\right\| \leq L \cdot H_{\max} \cdot \varepsilon$, where $\ell_b$ is the loss under RepL, $\ell$ is the loss under the base network, $H_{\max}$ captures the number of replacements, and $\varepsilon$ summarizes the local approximation error. We now provide a concrete measurement of this bias on the same ViT-tiny / CIFAR-10 setting.

**RepL model:** We use the trained RepL ViT-tiny checkpoint on CIFAR-10 corresponding to the "replace" setting in our main experiments:

depth 12, embedding dimension 192, 3 heads, patch size $8 \times 8$;

periodic removal with interval 4, excluding the last block. Under this schedule, there are three replacement sites at depth indices 2, 6 and 10 (0-based).

**Training setup (summary).** The RepL ViT-tiny is trained using the same pipeline as the baseline ViT:

optimizer: AdamW with weight decay 0.05;

initial learning rate: $1 \times 10^{-3}$ (cosine decay with 5-epoch linear warmup);

batch size: 512;

epochs: 250;

data augmentations identical to the baseline ViT.

**Experimental protocol.** In our implementation, each replacement site is realized by a lightweight computing layer that adds a synthesized residual update (constructed from neighboring blocks) to the hidden representation. For the bias diagnostic, we exploit the fact that the contribution of each computing layer can be scaled continuously; in particular, we can:

set the scale to 0 to effectively disable the replacement contribution at that site (only the skip path remains);

set the scale to 1 to fully enable the replacement contribution at that site.

Using this mechanism, we define:

$F$: the baseline network, obtained by disabling the replacement contribution at all computing layers (scales set to 0). This corresponds to using only the kept backbone blocks with their trained weights.

$F_b^{(r)}$: the same network where the first $r$ replacement sites (in depth order) are enabled (scale = 1), and the remaining ones are kept disabled (scale = 0), with $r \in \{0, 1, 2, 3\}$. All parameters of the backbone are shared between $F$ and $F_b^{(r)}$.

For each $r$, we measure two quantities:

**Forward deviation on logits:** $d_{\text{fwd}}(r) = \mathbb{E}_x \big[ \| F_b^{(r)}(x) - F(x) \|_2 \big]$, where the norm is taken over the class logits for each sample.

**Gradient deviation on shared parameters:** $d_{\text{grad}}(r) = \big\| \nabla_\theta \ell_b^{(r)} - \nabla_\theta \ell \big\|_2$, where $\ell$ and $\ell_b^{(r)}$ are the cross-entropy losses of $F$ and $F_b^{(r)}$, and $\theta$ includes all shared parameters (we explicitly exclude the parameters of the computing layers when forming the gradient vector). We also report the normalized ratio $\rho(r) = \frac{d_{\text{grad}}(r)}{\|\nabla_\theta \ell\|_2}$.

In practice, we estimate these quantities on CIFAR-10 test batches with:

20 batches (batch size 256) to estimate $d_{\text{fwd}}(r)$;

10 batches (batch size 256) to estimate $d_{\text{grad}}(r)$ and $\rho(r)$.

**Results.** The measured deviations are: We observe that:

| active replacements $r$ | $d\_\text{fwd}(r)$ (mean logit $\ell_2$) | $d_{\text{grad}}(r)$ | $\rho(r) = \frac{d_{\text{grad}}(r)}{\|\nabla_\theta \ell\|_2}$ |
|:---:|:---:|:---:|:---:|
| 0 | 0.000000 | 0.000000 | 0.0000 |
| 1 | 1.056977 | 1.467278 | 0.1431 |
| 2 | 1.093139 | 1.498927 | 0.1491 |
| 3 | 1.141932 | 1.545640 | 0.1597 |

1. At $r = 0$, we have $F_b^{(0)} = F$ by construction, so both forward and gradient deviations are exactly zero.

2. When we enable a single replacement site ($r = 1$), the normalized gradient bias is $\rho(1) \approx$ **0.143**, i.e., the difference between $\nabla_\theta \ell_b^{(1)}$ and $\nabla_\theta \ell$ is about 14% of the baseline gradient norm. This indicates a modest and controlled bias in the shared-parameter gradients.

3. When we enable two or three replacement sites ($r = 2, 3$), both the forward and gradient deviations increase slowly and smoothly: - $\rho(2) \approx$ **0.149**, - $\rho(3) \approx$ **0.160**. The growth from $r = 1$ to $r = 3$ is mild and close to linear in $r$, consistent with our non-expansive composition analysis involving $H_{\max}$.

Overall, these diagnostics show that the empirical bias $\varepsilon$ entering $\left\| \nabla_\theta \ell_b - \nabla_\theta \ell \right\| \leq L \cdot H_{\max} \cdot \varepsilon$ is small (with $\rho(r) < 0.16$ even when all replacement sites are enabled) and grows slowly as more blocks are replaced. This provides direct empirical support that RepL introduces a controlled and modest bias in both forward predictions and shared-parameter gradients in the regimes considered in our experiments.

