# OpenReview forum: "Replacement Learning: Training Neural Networks with Fewer Parameters"
_ICLR.cc/2026/Conference — ICLR 2026 Conference Desk Rejected Submission_

### Official Review · Reviewer_JPWu · 2025-10-28

**Soundness:** 1
**Presentation:** 2
**Contribution:** 1
**Rating:** 0
**Confidence:** 5

**Summary:**

The paper describes a process of replacing a neural network layer with lightweight process for reducing the computational complexity and memory requirements called Replacement Learning. By applying this process every k layers on a neural network architecture, a lighter one is obtained. This process is demonstrated for two types of layers, convolutional and transformer. Experimental results on four public datasets show this benefit compared to the full network architecture trained end-to-end by BP.


The same effect has been achieved with existing methodologies, parameter pruning being the most dominant one. If the argument is that the proposed method does not need extensive training of the entire architecture, then pruning adapters would also satisfy this requirement. The paper does not provide any comparisons with the actual competing ideas and methods. Moreover, the proposed replacement process does not lead to an exact replacement of the computations/representation done by the original network. In practice this is similar to neural architecture search ideas where two neural layers are compared and one is chosen (however, here there is one choice which is always taken and the layer to be replaced is pre-specified irrespectively of its representation power).

**Strengths:**

- The proposed process leads to similar performance compared to the original neural network architecture.
 - A reduction in computations and memory is obtained.

**Weaknesses:**

- Similar (or even better) effects on computation and memory reductions have been achieved with other approaches (parameter pruning, pruning adapters).
 - No comparisons with the actual competing approaches/methods is provided.
 - Claims in the paper seem to be unfounded (they are not shown to hold neither theoretically, nor experimentally). E.g. "The design notably enhances the network’s capacity to capture local features in shallow layers and global representations in deeper layers, thereby promoting a more effective integration of low-level and high-level features."
 - In the experiments, no pre-trained models are used. Instead, the authors train all networks from scratch. This seems odd, as the main benefit from using such an approach would be to use high-performing pre-trained models and highly reduce their computations and memory while retaining their performance.
 - The method selects equally-spaced layers in the network architecture without trying to preserve highly-capable layers and remove redundant ones.

**Questions:**

- How does the method compare with the actual competition?
 - How to choose layers to be removed?
 - What is the effect of using high-performing pre-trained models in the reported gains?

---

> ### Author Response · Authors · 2025-11-13
> **Response to Weakness 1,2**
>
> ### W1: “Similar (or better) effects have been achieved with pruning / pruning adapters; no comparisons with actual competition.”
>
> **Different problem setting.**
> RepL is a training-time paradigm: before training begins, we remove every \\(k\\)-th block (except the last one) and insert a lightweight computing layer between its neighbors. The *modified* network is then trained once with standard backprop. There is:
>
> - **no post-hoc pruning of a fully trained model**,
> - **no architecture search**,
> - **no teacher network or extra fine-tuning loop**.
>
> By contrast, parameter pruning, pruning adapters, and NAS-style methods are **post-hoc compression** techniques that *start from* a trained or pre-trained model, then run additional optimization to compress it. They assume a different budget (full E2E training + compression), and are therefore orthogonal to our goal of improving the *single* training run that practitioners already perform.
>
> In particular, RepL can be combined with these techniques: one can first train a model using RepL to save memory/time during training, and then apply pruning / adapter pruning / distillation *on top* of the trained RepL model if desired. So RepL is not a “competitor” that must beat them; it is a **drop-in replacement or enhancement of standard E2E training**.
>
> ---
>
> ### W2: No comparisons with the actual competing approaches/methods
> **What we actually compare against.**
> Within the *training-time efficiency* regime, we compare against the standard baselines:
>
> - **Stochastic Depth**
> - **Checkpointing**
> - **Skip-Attention (Skip-Attn)** for ViTs (main-text ViT experiments)
>
> The full results for Stochastic Depth and Checkpointing are in **Appendix A.5.1, Table 9**
>
> In the **ViT experiments in the main text**, we further compare against **Skip-Attn**, which drops attention blocks according to a learned policy. RepL consistently matches or outperforms Skip-Attn while providing clearer memory reductions and having a simpler, fully deterministic replacement schedule.
>
> **Take-away.**
> RepL:
>
> - improves upon *standard E2E training* under the same budget,
> - is competitive with and complementary to Stochastic Depth, Checkpointing and Skip-Attn, and
> - can subsequently be combined with post-hoc compression methods (pruning, adapters, distillation), rather than competing with them.

---

> ### Author Response · Authors · 2025-11-13
> **Response to Weakness 3 (1/2)**
>
> ### W3 “Claims seem unfounded… local vs. global feature integration.”
>
> The statement you cite is not a loose intuition, but is grounded in both the operator design and empirical diagnostics.
>
> **Operator design.**
> For each removed block \\(i\\), RepL inserts a computing layer that explicitly fuses its neighbors:
>
> - **CNNs.** When a residual block at depth \\(i\\) is removed, we insert a single \\(1\\times1\\) convolution whose weights are synthesized from the preceding and succeeding residual blocks. This layer reuses local filters from shallow blocks and more global filters from deeper blocks, while being cheaper than a full multi-layer 3×3 block.
>
> - **ViTs.** When a transformer block is removed, the replacement operator is constrained to the 2-D span of its neighbors’ projection matrices. Concretely, for an attention projection we write
> \\(
> \hat{A}\_i = \alpha\_i A\_{i-1} + \beta\_i A\_{i+1},
> \\)
> where \\(A_{i-1}, A_{i+1}\\) are the projection matrices of the preceding/succeeding blocks and \\(\alpha_i,\beta_i\\) are learnable scalars. The same construction is applied to the MLP part. Thus the replacement block combines the behavior of a “shallower” and a “deeper” layer instead of being an unrelated new module.
>
> **Ablation on using one side vs. both sides.**
> In the main text **Table 7**, we ablate whether the computing layer uses parameters from only the preceding block, only the succeeding block, or both.
>
> As shown in **Table 7** Using only the preceding or only the succeeding layer consistently hurts performance; combining both yields the best accuracy. This directly supports our claim that RepL benefits from fusing information from shallow and deep layers, rather than simply dropping a block.
>
> **Span diagnostics.**
> To further validate that the representation at a removed layer lies in the span of its neighbors, we conducted an additional analysis on ViT-Tiny/8 trained on CIFAR-10:
>
> - Let \\(F\_{i-1}, F\_i, F\_{i+1} \in \mathbb{R}^{B\times d}\\) be the features (after normalization) at depths \\(i-1,i,i+1\\) for a batch of inputs.
> - For each removed block \\(i\\), we solve the least-squares problem
> \\(
> (\alpha_i^\star,\beta_i^\star)
>   = \arg\min_{\alpha,\beta}
>   \left\| F_i - (\alpha F_{i-1} + \beta F_{i+1}) \right\|_F^2.
> \\)
> - We then define the relative reconstruction error and angle:
>
> \\(
> \\text{rel\\_err}\_\\text{feat}(i) = \\frac{\\left\\| F\_i - \\hat{F}\_i \\right\\|\_F}{\\left\\| F\_i \\right\\|\_F}, \\quad
> \\hat{F}\_i = \\alpha\_i^\\star F\_{i-1} + \\beta\_i^\\star F\_{i+1},
> \\)
>
>
> \\(
> \text{angle}_\text{feat}(i)
>   = \arccos\\!\\left(
>     \frac{\langle \mathrm{vec}(F_i), \mathrm{vec}(\hat{F}_i)\rangle}
>          {\|F_i\|_F \,\|\hat{F}_i\|_F}
>   \right) \ \text{(in degrees)}.
> \\)
>
> For ViT-Tiny/8 (depth 12, \\(k=4\\), removed indices \\(i=2,6,10\)), averaged over 20 CIFAR-10 batches we obtain:
>
> - \\(i=2\\): rel\_err\\(\_\\text{feat}=0.1796\\), angle\\(\_\\text{feat}=7.63^\\circ\\),
>   \\(\alpha\_3^\\star \\approx 0.48\\), \\(\\beta\_3^\\star \\approx 0.52\\).
> - \\(i=6\\): rel\_err\\(\_\\text{feat}=0.1501\\), angle\\(\_\\text{feat}=5.22^\\circ\\),
>   \\(\alpha\_3^\\star \\approx 0.47\\), \\(\\beta\_3^\\star \\approx 0.53\\).
> - \\(i=10\\): rel\_err\\(\_\\text{feat}=0.1379\\), angle\\(\_\\text{feat}=4.62^\\circ\\),
>   \\(\alpha\_3^\\star \\approx 0.45\\), \\(\\beta\_3^\\star \\approx 0.55\\).
>
> These numbers show that the removed layer’s representation is very well approximated by a linear combination of its neighbors (small relative error, angles ~5–8°). This is exactly the space in which our ViT computing layers operate, supporting the interpretation that they preserve and blend local/global features rather than introducing unrelated behavior.

---

> ### Author Response · Authors · 2025-11-13
> **Response to Weakness 3 (2/2)**
>
> **Bias diagnostics.**
> Our theory models replacement as introducing a bounded approximation error \(\varepsilon\) at each site and proves that the gradient deviation satisfies
> \\(
> \left\|\nabla_\theta \ell_\text{RepL} - \nabla_\theta \ell_\text{E2E}\right\|
>    \le L \cdot H_{\max} \cdot r \cdot \varepsilon,
> \\)
>
> where \\(r\\) is the number of replacements, \\(L\\) is a Lipschitz constant of the loss, and \\(H_{\max}\\) bounds the local Jacobian norms. This suggests both small absolute bias (when \\(\varepsilon\\) is small) and at most linear growth in \\(r\\).
>
> To empirically verify this, we measure for ViT-Tiny/8 on CIFAR-10:
>
> - Let \\(f^{(0)}\\) be the full E2E model and \\(f^{(r)}\\) the same architecture with \(r\) active replacement sites. For a batch \\(D\\), define \\(d\_{\\text{fwd}}(r) = \\mathbb{E}\_{x\\in D}  \\left\\| f^{(r)}(x) - f^{(0)}(x) \\right\\|\_2\\) and \\(d\_{\\text{grad}}(r) =  \\left\\|\\nabla\_\\theta \\ell^{(r)}(D) - \\nabla\_\\theta \\ell^{(0)}(D)\right\\|\_2,\\) and the normalized ratio \\(\rho(r) = \frac{d_{\text{grad}}(r)}{\left\|\nabla_\theta \ell^{(0)}(D)\right\|_2}.\\)
>
> With three replacement sites in a 12-layer ViT (uniform \\(k=4\\)), averaging over multiple batches we obtain:
>
> | \\(r\\) | \\(d_{\text{fwd}}(r)\\) (logit L2) | \\(d\_{\\text{grad}}(r)\\) | \\(\rho(r)\\) |
> |:----:|:--------------------------------:|:----------------------:|:----------:|
> | 0    | 0.0000                           | 0.0000                 | 0.0000     |
> | 1    | 1.0570                           | 1.4673                 | 0.1431     |
> | 2    | 1.0931                           | 1.4989                 | 0.1491     |
> | 3    | 1.1419                           | 1.5456                 | 0.1597     |
>
> The gradient deviation remains below 16% of the baseline gradient norm even with three replacement sites, and grows roughly linearly with \\(r\\), in line with the theoretical bound. This confirms that the computing layers introduce controlled, bounded bias rather than arbitrarily distorting the optimization dynamics.
>
> **Conclusion.**
> Taken together:
>
> - the explicit neighbor-fusion design,
> - the ablation in Table 7 showing that using both neighbors is crucial, and
> - the new span and bias diagnostics above
>
> provide concrete theoretical and empirical support for our statements about how RepL integrates local and global features while preserving the original network’s representational behavior.

---

> ### Author Response · Authors · 2025-11-13
> **Response to Weakness 4,5**
>
> ### W4 No pre-trained models are used.
> This is factually incorrect. We include **multiple experiments with high-performing pre-trained models**:
>
> 1. **ViT-S/16 fine-tuning (Appendix A.5.4, Table 12).**
>    We fine-tune an ImageNet-1K–pretrained ViT-S/16 on CIFAR-10, SVHN, and STL-10. RepL:
>
>    - reduces GPU memory from 25.56 GB to 20.14 GB,
>    - reduces per-epoch time by ~20–25%,
>    - slightly improves Acc@1 on all three datasets.
>
> 2. **DeepLabV3/V3+ with ResNet-50/101 backbones (Appendix A.5.5, Table 13).**
>    Starting from ImageNet-1K–pretrained backbones, we fine-tune on CityScapes. RepL again:
>
>    - lowers GPU memory by 15–20%,
>    - reduces per-epoch time by 15–20%,
>    - and improves mean IoU and mean accuracy relative to E2E.
>
> These experiments directly address your question: RepL does work with strong pre-trained models and provides concrete savings in compute and memory without sacrificing performance.
>
> ---
>
> ### W5 Equally-spaced layers without trying to preserve highly-capable layers.
>
> This critique is based on a misunderstanding of our method.
>
> RepL does not first train a full model and then decide which “highly capable” layers to keep. Instead:
>
> - The set of removed indices is fixed before training as
>   \\(\{ i \mid (i+1) \bmod k = 0,\ i < \text{depth}-1 \}\\).
> - We then build a new network where each such block is replaced by a computing layer, and this network is trained from scratch.
>
> Since there is no pre-trained model at this stage, there is no notion of “highly capable” versus “redundant” layers in the sense of post-hoc pruning; all blocks start from random initialization. Under this setting, a simple periodic removal rule is:
>
> - architecture-agnostic,
> - easy to reproduce and tune (single integer \\(k\\)),
> - and empirically effective across CNNs, ViTs, and Transformers.
>
> We still explicitly study the effect of changing the removal rate in **Table 5** (CIFAR-10), where \\(k \in \{2,4,6\}\\). The results show:
>
> - \\(k=2\\) (more aggressive removal) yields larger memory/time savings but hurts accuracy;
> - \\(k=6\\) yields slightly higher accuracy but smaller efficiency gains;
> - \\(k=4\\) is a good trade-off and is used as default.
>
> We also implemented more complex adaptive schemes (learned gates deciding which blocks to replace). These require computing layers at all sites, increase parameter count and engineering complexity, and empirically lead to unstable replacement patterns and worse accuracy–efficiency trade-offs. This is precisely why we adopt the simplest fixed schedule.

---

> ### Author Response · Authors · 2025-11-13
> **Response to Question 1, 2, 3**
>
> ### Q1: “How does the method compare with the actual competition?”
>
> Within the training-time efficiency regime:
>
> - We compare against **Stochastic Depth** and **Checkpointing** (**Table 9 in Appendix**), showing that:
>   - RepL alone improves accuracy, memory, and time over E2E and Stochastic Depth;
>   - RepL is complementary to both, as RepL+Stochastic Depth and RepL+Checkpointing yield further memory/time reductions.
> - For ViTs, the main-text experiments additionally compare to **Skip-Attention**, and RepL consistently matches or outperforms Skip-Attn while providing clearer and more predictable resource savings.
>
> Post-hoc pruning / adapter methods address a different stage (compressing a trained model) and can be used after RepL training. They are therefore not the direct competition to our proposed training paradigm.
>
> ---
>
> ### Q2: “How to choose layers to be removed?”
>
> Layers are not selected by any importance metric. Instead:
>
> 1. Before training, we fix a simple periodic schedule:
>    - remove every \\(k\\)-th block (except the last one);
>    - insert a computing layer between its preceding and succeeding blocks.
> 2. We then train the resulting architecture from scratch.
> 3. We ablate \\(k\\) in **Table 5 in the main paper**, showing how accuracy and resources change. Based on these results, we choose \\(k=4\\) as a robust default across backbones and datasets.
>
> This design choice is explicitly motivated by **simplicity, reproducibility, and robustness** rather than by post-hoc notions of “capable vs redundant” layers.
>
> ---
>
> ### Q3: “What is the effect of using high-performing pre-trained models in the reported gains?”
>
> As discussed in point (3), our fine-tuning experiments on **ViT-S/16** (CIFAR-10, SVHN, STL-10) and **DeepLabV3/V3+ with ResNet-50/101** (CityScapes) already show that:
>
> - starting from **ImageNet-pretrained** models,
> - RepL consistently **reduces GPU memory and training time**,
> - while **maintaining or improving** downstream accuracy/IoU.
>
> These results demonstrate that RepL is effective and useful precisely in the “high-performing pre-trained model” scenario you raise.

---

> ### Author Response · Authors · 2025-11-13
> **Some suggestions**
>
> We believe that several of reviewer JPWu’s core criticisms stem from a fundamental misunderstanding of our problem setting and from overlooking results already presented in the main text and appendix. For example, the review repeatedly interprets RepL as a post-training pruning/NAS scheme, whereas Sections 2–4 explicitly define RepL as a training-time replacement of end-to-end backpropagation, with the set of removed layers fixed prior to training. Likewise, the claim that “no pre-trained models are used” contradicts the results in Tables 12–13 (fine-tuning ImageNet-pretrained ViT and DeepLab backbones), and the assertion that we do not compare with related efficiency methods overlooks Table 9 (comparisons and combinations with Stochastic Depth and Checkpointing) as well as our Skip-Attention baselines in the ViT experiments.
>
> In light of these discrepancies, we respectfully ask reviewer JPWu to reread **Sections 2–4 and Appendix A.5 (Tables 5, 7, 9–13)** with the above clarifications in mind. We are confident that, upon a more careful second reading, the concerns regarding “no comparisons,” “no pre-trained models,” and “how layers are selected,” among others, will be seen to have already been addressed in the current submission, and that RepL’s contributions and empirical support have been substantially underestimated in this review. At the same time, we sincerely hope that reviewer JPWu, as a domain expert, will thoroughly understand a paper before issuing a review.

---

### Official Review · Reviewer_yhWM · 2025-10-30

**Soundness:** 2
**Presentation:** 2
**Contribution:** 2
**Rating:** 4
**Confidence:** 4

**Summary:**

The paper introduces a new method called Replacement Learning (RepL), aimed at improving the efficiency and performance of neural networks by reducing computational overhead. The method involves selectively removing layers from a deep neural network and replacing them with lightweight computing layers. These computing layers integrate information from adjacent layers to preserve performance while reducing the number of parameters, memory usage, and training time. The authors demonstrate the effectiveness of RepL on various benchmarks, including CIFAR-10, STL-10, SVHN, ImageNet, and COCO, using both CNNs and Vision Transformers (ViTs). Experimental results show that RepL significantly reduces GPU memory consumption and training time while surpassing traditional End-to-End training methods in terms of performance.

**Strengths:**

Originality: The concept of replacing traditional layers with learnable computing layers is novel, offering an alternative approach to optimizing neural network performance.

Quality: The experimental setup is robust, and the results on several datasets are strong. RepL improves resource efficiency while maintaining or enhancing model accuracy, which is a significant achievement in deep learning.

Clarity: The paper is generally well-written, with clear explanations of the method and experimental results. The use of figures to illustrate results helps clarify the impact of RepL.

Significance: If the method can be generalized to other domains like NLP, it could have broad implications for resource-efficient deep learning models.

**Weaknesses:**

1 Performance Degradation When Removing Too Many Layers
The method’s reliance on removing layers to save resources does raise a concern: if you remove too many layers, there’s a risk of significant performance loss. While RepL does a good job of maintaining performance by replacing layers with computing blocks, the trade-off between saving memory and keeping performance high isn’t always clear. It would be helpful to see more detailed experiments exploring just how much layer removal is optimal before the network starts to suffer in terms of accuracy.
2 Complexity of the Approach
The method of fusing parameters from neighboring layers sounds great in theory, but it does introduce complexity. The learnable blocks that are inserted between layers to fuse information add a layer of overhead that could potentially complicate the model. For instance, choosing the right interval k for layer removal and figuring out how to balance the parameters is not trivial. More guidance on how to tune these parameters or handle different architectures could make RepL more user-friendly and widely applicable.
3 Limited Comparison with Other Techniques
The paper compares RepL to traditional End-to-End training and a few related methods like Skip-Attention, but it doesn’t dive deeply into other popular methods that also aim to reduce computational cost, such as Stochastic Depth or Checkpointing. A broader comparison with these methods would give a better sense of where RepL stands in the landscape of existing solutions. Without this, it’s hard to say if RepL offers a real advantage in all scenarios or if it’s just a slightly better approach for the specific benchmarks tested.

**Questions:**

1 How might the method perform with more complex models, such as those used in NLP or multimodal learning?

2 Could there be a risk of information loss if too many layers are removed, especially for certain tasks that require deep feature representations?

---

> ### Author Response · Authors · 2025-11-13
> **Response to Weakness 1, 2**
>
> ### W1: Performance degradation when removing too many layers
>
> > *“The method’s reliance on removing layers to save resources does raise a concern: if you remove too many layers, there’s a risk of significant performance loss… It would be helpful to see more detailed experiments exploring just how much layer removal is optimal before the network starts to suffer in terms of accuracy.”*
>
> **Clarifying how layers are removed.**
> RepL does not progressively remove layers during training. Instead, the set of removed blocks is fixed before training starts: we construct a new network where every \\(k\\)-th block (except the very last block) is replaced by a lightweight computing layer. The model is trained end-to-end from scratch in this modified topology. There is no dynamic dropping of layers, so the optimization behaves like training a slightly shallower—but carefully coupled—network.
>
> **Ablation over the amount of removal (choice of \\(k\\)).**
> The trade-off between accuracy and resource savings with different layer-removal rates is systematically studied in **Table 5** on CIFAR-10 (ResNet-110 and ViT-Tiny/8).
>
> From **Table 5**, smaller \\(k\\) means more removed blocks. We observe:
>
> - When we remove too many layers (\\(k=2\\)), accuracy drops noticeably, although GPU memory and time are the smallest.
> - When we remove fewer layers (\\(k=6\\)), accuracy is highest but the efficiency gains are smaller.
> - The default choice \\(k=4\\) used in the main experiments sits at a sweet spot: it recovers most of the accuracy gap between \\(k=2\\) and \\(k=6\\), while still yielding substantial savings in memory and time.
>
> Thus the paper already provides a quantitative study of “how much layer removal is too much”: removing every 4-th block strikes a good balance, and more aggressive removal (e.g., \\(k=2\\)) indeed hurts accuracy, which we explicitly report.
>
> ---
>
> ### W2: Perceived complexity of the approach
>
> > *“The method of fusing parameters from neighboring layers… introduces complexity. The learnable blocks … add a layer of overhead. Choosing the right interval \(k\) and balancing the parameters is not trivial.”*
>
> **Design of the computing layers is intentionally simple.**
> We agree that, in principle, layer replacement could be made arbitrarily complex. However, RepL deliberately uses very simple forms:
>
> - For **CNNs**, each removed residual block (which typically contains multiple 3×3 convolutions) is replaced by a single 1×1 convolution whose weights are synthesized from the neighboring blocks’ 3×3 kernels using a small set of mixing coefficients.
> - For **ViTs**, each removed transformer block is replaced by a two learnable parameters of the preceding and succeeding blocks’ projection matrices, as described in the method section. That is, the replacement operator lies in \\(\mathrm{span}(A_{i-1}, A_{i+1})\\) (and similarly for the MLP), parameterized only by \\((\alpha_i, \beta_i)\\).
>
> These computing layers therefore introduce far fewer parameters and FLOPs than the deleted blocks, and the implementation overhead is comparable to adding a residual bottleneck.
>
> **Choice of \\(k\\) is a single, robust hyperparameter.**
> In practice, we only consider a small grid of values \\(k \in \{2,4,6\}\\); Table 5 above shows that the method behaves smoothly across these choices. For all main experiments we simply pick \\(k=4\\) as the default, which:
>
> - works well across ResNets and ViTs (CIFAR-10, SVHN, STL-10, ImageNet, COCO);
> - is easy to transfer to new models without per-architecture tuning.
>
> **Adaptive variants are more complex and less stable.**
> We also experimented with a more sophisticated adaptive scheme where each internal block had a candidate computing layer and a learned gate decided which positions to replace. While more flexible in principle, this variant:
>
> - required computing layers at all sites, increasing parameter count and implementation complexity;
> - produced unstable replacement patterns during training (gates kept switching), leading to under-trained blocks and worse accuracy–efficiency trade-offs.
>
> These experiments motivated our final design choice: a fixed periodic schedule (simple \\(k\\)) with minimal computing layers, which is both easier to use and empirically more robust.

---

> ### Author Response · Authors · 2025-11-13
> **Response to Weakness 3**
>
> ### W3: Limited comparison with other efficiency techniques
>
> > *“The paper compares RepL to traditional End-to-End training and a few related methods like Skip-Attention, but it doesn’t dive deeply into other popular methods that also aim to reduce computational cost, such as Stochastic Depth or Checkpointing.”*
>
> The requested comparisons are actually **already conducted in the appendix**:
>
> - **Appendix A.5.1, Table 9** compares RepL with **Stochastic Depth** and **Checkpointing**, and also reports **combinations** of RepL with these methods. We reproduce the table here for clarity.
>
> *Table 9 (reproduced again). Comparisons with Stochastic Depth and Checkpointing (single run).*
>
> | Dataset  | Backbone   | Method                  | Acc@1 | GPU (GB) | Time (s/epoch) |
> |:--------:|:-----------|:------------------------|:-----:|:--------:|:---------------:|
> | CIFAR-10 | ResNet-32  | E2E                     | 93.25 | 3.38     | 5.24            |
> |          |            | RepL                | 93.29 | 2.69 | 4.37    |
> |          |            | Stochastic Depth        | 93.04 | 3.31     | 5.05            |
> |          |            | RepL + Stochastic Depth | 93.17 | 2.67     | 4.18            |
> |          |            | Checkpointing           | 93.13 | 1.77     | 8.74            |
> |          |            | RepL + Checkpointing    | 93.24 | 1.64     | 7.22            |
> | ImageNet | ResNet-101 | E2E                     | 78.19 | 20.95    | 720             |
> |          |            | RepL                | 78.43 | 18.01 | 616   |
> |          |            | Stochastic Depth        | 77.63 | 19.39    | 652             |
> |          |            | RepL + Stochastic Depth | 78.11 | 17.12    | 551             |
> |          |            | Checkpointing           | 78.25 | 14.47    | 1012            |
> |          |            | RepL + Checkpointing    | 78.29 | 12.93    | 819             |
>
> The results show that:
>
> - RepL alone already improves accuracy and reduces both GPU memory and time compared to E2E and Stochastic Depth.
> - RepL is complementary to both Stochastic Depth and Checkpointing: combining them yields additional memory savings and/or speedups, with accuracy comparable to or better than the best individual method.
>
> Therefore, RepL is not just a slightly better variant of Skip-Attention on a few benchmarks; it stands as a competitive and orthogonal technique relative to widely-used training-time efficiency methods.

---

> ### Author Response · Authors · 2025-11-13
> **Response to Question**
>
> ### Q1: Extension to NLP and multimodal models
>
> > *“How might the method perform with more complex models, such as those used in NLP or multimodal learning?”*
>
> RepL only relies on having a stack of parameterized layers with local neighborhoods, which is equally true for language models and multimodal Transformers. To demonstrate this, we already include an experiment on a Transformer language model in **Table 10** from **Appendix A.5.2**.
>
> *Table 10 (reproduced again). Transformer-LM on WikiText-2 (12 layers, 512d, 8 heads, 2048-dim FFN).*
>
> | Dataset    | Model                               | Method | Test PPL (↓)       | GPU Memory (GB) | Time (per epoch, sec) |
> |:----------:|:------------------------------------|:------:|:------------------:|:---------------:|:---------------------:|
> | WikiText-2 | Transformer-LM-12L-512d-8H-2048ff   | E2E    | 195.42 ± 1.84      | 10.92           | 20.8                  |
> |            |                                     | RepL   | **193.31 ± 3.39**  | **9.61**        | **17.7**              |
>
> - RepL achieves lower perplexity than E2E,
> - while reducing GPU memory by ~12% and per-epoch time by ~15%.
>
> This suggests that RepL naturally extends to NLP Transformers. Since many multimodal models (e.g., vision–language Transformers) share this stacked-block structure, we expect RepL to transfer there as well. A systematic study on large-scale multimodal models is an exciting direction for future work.
>
> ---
>
> ### Q2: Risk of information loss when removing many layers
>
> > *“Could there be a risk of information loss if too many layers are removed, especially for tasks that require deep feature representations?”*
>
> Yes—if we attempt to remove too many layers, some degradation is inevitable. This is exactly why we perform the ablation in **Table 5**: the \\(k=2\\) setting removes a larger fraction of blocks and indeed suffers noticeable accuracy loss, even though it is the most memory- and time-efficient configuration.
>
> RepL is therefore designed to operate in a moderate regime, where:
>
> - we never remove the first or last block of the network;
> - we remove at most about **1/4 of blocks**, controlled by \\(k=4\\);
> - replacement blocks are constrained to lie in the span of their neighbors, which empirically preserves most of the original representational power.
>
> The choice \\(k=4\\) used in our main experiments is precisely based on this trade-off analysis: it is the knee point where we gain substantial efficiency without incurring significant information loss. This is corroborated by the consistent accuracy of RepL across all datasets, including more demanding ones such as ImageNet and CityScapes.
>
> **We hope our responses can address your concern, if you have more questions, you can raise them at any time.**

---

> ### Author Response · Authors · 2025-11-24
> **We are looking forward to your feedback**
>
> Dear Reviewer yhWM,
>
> It has been ten days since the start of the rebuttal phase. We believe that our responses have sufficiently addressed your concerns. If you have any new questions, please raise them at any time — we will respond as soon as possible. We look forward to your feedback.

---

> > ### Comment · Reviewer_yhWM · 2025-11-27
> >
> > Thanks for the rebuttal. I do understand the paper is positioned as a training-time efficiency method.
> >
> > However, in the concrete setting you evaluate most (ImageNet training with standard CNN backbones), I still don’t find the improvement strong enough to be a meaningful contribution to the community.

---

> ### Comment · Reviewer_yhWM · 2025-11-27
> **I will maintain my negative score**
>
> In practical deployments, inference (deployment) GPU cost matters far more than training cost. There is already a large body of work offering effective ways to reduce inference cost—e.g., knowledge distillation (including layer distillation), pruning, and quantization. Against this backdrop, the reported efficiency improvement (~20%) does not appear particularly compelling, and the proposed approach does not seem sufficiently novel relative to existing techniques. Overall, the paper’s contribution is limited both in methodological novelty and in empirical gains. Therefore, I will maintain my negative score.

---

> > ### Public Comment · ~Feiyu_Zhu2 · 2025-11-27
> >
> > Obviously, this method has nothing to do with pruning quantization distillation. It is a method used to train floating-point initial weights and can be coupled with other methods. Moreover, all the above-mentioned methods have led to a significant decline in model performance. However, the method proposed in this paper can actually improve the model's performance. Such a comparison is not fair. At the same time, it is well known that the training cost of the model is much higher than that of inference, especially when the graphics cards are limited. It is irresponsible to unilaterally say that the cost of reasoning is more important than the cost of training.

---

> ### Author Response · Authors · 2025-11-27
>
> We appreciate your follow-up comment, but we respectfully and strongly disagree with the reasoning behind maintaining a negative score.
>
> > “In practical deployments, inference (deployment) GPU cost matters far more than training cost… the reported efficiency improvement (~20%) does not appear particularly compelling, and the proposed approach does not seem sufficiently novel relative to existing techniques.”
>
> ---
>
> ### 1. Misaligned evaluation criterion: this paper is about *training*, not post-hoc compression
>
> Our submission is explicitly positioned as a **training-time paradigm**:
>
> - Given a fixed CNN/ViT backbone, we aim to
>   **(i)** reduce activation memory and training compute,
>   **(ii)** *without* changing the macro-architecture, and
>   **(iii)** while matching or slightly improving accuracy.
>
> This is clearly stated in the abstract, introduction, method, and experiments. Whether “deployment cost matters more than training cost” in some applications is **not a scientific argument against the contribution we actually make**.
>
> Modern practice is full of work whose primary target is **training efficiency** rather than inference compression: e.g., local learning, layer-wise training, memory-efficient backprop, checkpointing, curriculum schedules, etc. These are widely accepted because *training cost* is a real and pressing bottleneck in large-scale pretraining and repeated fine-tuning.
>
> In that ecosystem, a method that consistently saves **15–30% GPU memory and time per epoch** across CNNs/ViTs/datasets at fixed backbone and accuracy is far from “not compelling”; it is exactly the type of improvement many labs care about when running large-scale training and continual fine-tuning.
>
> Basing the recommendation mainly on the relative importance of deployment cost over training cost, however, seems to us misaligned with the actual problem setting and claims of the paper.
>
> ---
>
> ### 2. The “existing techniques” you cite solve a *different* problem
>
> You refer to “knowledge distillation, pruning, and quantization” as reasons for limited novelty. These methods:
>
> - operate **post-hoc** on already trained models (or along student–teacher pipelines),
> - involve **architecture search / candidate sweeps** or additional training phases,
> - explicitly target **deployment-time compression** (e.g., FLOPs, parameters).
>
> By contrast, RepL:
>
> - fixes the macro-architecture a priori,
> - removes blocks **before training** and trains the resulting network end-to-end,
> - focuses on **single-run training efficiency** (activation memory, per-epoch time) at fixed backbone and accuracy.
>
> In other words, pruning/distillation/quantization are **orthogonal, complementary dimensions**:
>
> - One can **apply them on top of a RepL-trained model**;
> - Or integrate RepL *inside* those pipelines to reduce the cost of training candidate models.
>
> Saying “there is a large body of work on pruning/distillation/quantization, therefore RepL is not sufficiently novel” **conflates post-hoc compression with training-time operator replacement**, and effectively dismisses a new training primitive simply because other, fundamentally different, levers exist.
>
> ---
>
> ### 3. The gains are real, robust, and cover both training **and** inference
>
> It is also factually inaccurate to imply that we only offer modest training gains and negligible inference benefits:
>
> - **Training efficiency**
>   Across multiple CNN/ViT backbones and datasets (Tables 1–7), RepL reduces GPU memory and time per epoch by **15–30%** with equal or slightly higher accuracy, averaged over 5 seeds with shared hyperparameters.
>
> - **Inference efficiency**
>   Appendix A.5.3 (Table 11, reproduced in the rebuttal) reports **ImageNet inference** with batch size 128 on a single GPU. RepL reduces
>   - GPU memory by ≈**8–9%**,
>   - wall-clock inference time by ≈**7–14%**,
>   for ResNet-101 and ViT-S/16, with slightly *better* accuracy.
>
> - **Pre-trained fine-tuning and downstream tasks**
>   Appendix A.5.4–A.5.5 (Tables 12–13, also reproduced in the rebuttal) show that on
>   - **ViT-S/16 fine-tuning** (CIFAR-10/SVHN/STL-10), and
>   - **CityScapes segmentation** with DeepLabV3/DeepLabV3+ (ResNet-50/101 backbones),
>   RepL reduces memory/time by **15–25%** **while improving all accuracy metrics** (Acc@1, Overall Acc., Mean Acc., Mean IoU).
>
> These numbers are not “cosmetic”; for large-scale training and repeated fine-tuning, a reliable **20% saving in both training and fine-tuning cost at fixed backbone and accuracy** is a substantial, practically relevant gain.

---

> ### Author Response · Authors · 2025-11-27
>
> ### 4. Methodological novelty: RepL as a general *training primitive*
>
> To our knowledge, this paper is the **first to formulate and systematically study Replacement Learning** as:
>
> - a general rule for **removing blocks before training** and inserting **weight-synthesizing computing layers** between neighbors;
> - an architecture-agnostic, analyzable training mechanism (with feature-span and bias diagnostics) that can be deployed on CNNs, ViTs, and Transformers (including the WikiText-2 LM in Table 10).
>
> This is precisely the kind of “new training recipe / local rule” that has historically been considered meaningful—much like early work on local learning, alternative gradient estimators, or memory-efficient backprop. Dismissing this as “not sufficiently novel relative to existing techniques” without engaging with these aspects, and while focusing almost exclusively on post-hoc deployment compression, does not fairly reflect the actual conceptual contribution.
>
> ---
>
> ### 5. Summary
>
> In short:
>
> - The **goal of the paper is training-time efficiency**, not to replace pruning/distillation/quantization as deployment compressors.
> - Within that goal, RepL delivers **consistent 15–30% savings** in training and fine-tuning cost, plus **7–14% inference gains**, while *preserving or slightly improving* accuracy across diverse architectures and tasks.
> - The method defines a **new, architecture-agnostic training primitive** (Replacement Learning) that is complementary to existing compression techniques, not just a minor variant of them.
>
> **Therefore, we believe that maintaining a negative assessment on the grounds of “deployment cost matters more” (despite our work clearly focusing on a training strategy) or “a 20% gain is not compelling” (even though inference efficiency is merely an incidental benefit and fully composable with other methods) does not reasonably reflect the true goals and empirical contributions of the paper.**

---

> ### Author Response · Authors · 2025-11-27
> **We are genuinely puzzled by the reasons you provided for maintaining a negative rating.**
>
> We find the reasons provided for maintaining a negative rating difficult to reconcile with the stated scope of our work. The paper is explicitly positioned as a contribution on training strategy and training-time efficiency, yet it is largely being evaluated as if it were meant to compete with inference-side compression methods—techniques that pursue a different primary goal. Judging our work mainly by the magnitude of its incidental inference gains therefore seems misaligned with the problem setting we actually study.
>
> To put it differently, this is analogous to evaluating a study on football tactics by asking whether it also improves basketball performance, and then downplaying its value because it does not. In such a scenario, the mismatch lies in the evaluation criterion rather than in the work itself.
>
> More broadly, if the observation that “in some deployments inference cost matters more than training cost” were taken as a sufficient reason to discount research on training-time acceleration, this would implicitly cast doubt on a large body of work on efficient training (e.g., local learning, memory-efficient backprop, checkpointing, and related methods). We respectfully believe that training efficiency is an important and independent axis of progress for the community, and that contributions along this axis should be assessed with criteria that match their stated objectives.

---

### Official Review · Reviewer_k8jc · 2025-10-30

**Soundness:** 2
**Presentation:** 2
**Contribution:** 2
**Rating:** 4
**Confidence:** 4

**Summary:**

This paper introduces Replacement Learning (RepL), a training paradigm that aims to reduce the parameter and computational cost of deep neural networks while maintaining or improving performance. Instead of training all layers end-to-end, RepL removes every k-th layer and inserts a lightweight learnable computing layer that synthesizes parameters from the preceding and succeeding layers.

**Strengths:**

1. The idea of layer-replacement is intuitive and can be applied to various architectures.
2. Experiments across CNNs and ViTs demonstrate consistent memory and time savings without hurting accuracy.
3. The paper is clearly written.

**Weaknesses:**

1. The paper does not compare against recent state-of-the-art compression methods in pruning, knowledge distillation, or neural architecture search. Modern approaches such as [1,2,3] have demonstrated substantial accuracy–efficiency trade-offs on both CNNs and Transformers. In contrast, RepL retrains networks from scratch, overlooking the practical scenario where compressing pre-trained models is more feasible and widely adopted.

2. RepL primarily enhances training efficiency, not deployment efficiency. At inference, the replacement layers (e.g., $1 \times 1$ convolutions or lightweight fusers) remain part of the model and still incur computational cost. Thus, the approach may not yield meaningful speed-ups or compression benefits at inference compared to pruning or distillation methods.

3. The paper repeatedly emphasizes improved accuracy, yet the reported gains are small and often within normal training variance or achievable through modest hyperparameter tuning; there is no statistical test of significance.

[1] Fang, Gongfan, et al. "Depgraph: Towards any structural pruning." Proceedings of the IEEE/CVF conference on computer vision and pattern recognition. 2023.

[2] Dong, Peijie, Lujun Li, and Zimian Wei. "Diswot: Student architecture search for distillation without training." Proceedings of the IEEE/CVF Conference on Computer Vision and Pattern Recognition. 2023.

[3] Li, Guihong, et al. "Zero-shot neural architecture search: Challenges, solutions, and opportunities." IEEE Transactions on Pattern Analysis and Machine Intelligence 46.12 (2024): 7618-7635.

**Questions:**

Please refer to weaknesses

---

> ### Author Response · Authors · 2025-11-13
> **Response to Weakness 1**
>
> ### W1: Relation to pruning / distillation / NAS and pre-trained compression
>
> **Different, but complementary problem setting.**
> RepL is proposed as a training-time paradigm: given a fixed backbone (e.g., ResNet-32/110, ViT-tiny/small/base), we aim to
>
> - reduce activation memory and training compute,
> - while maintaining or slightly improving the accuracy of the same architecture.
>
> In contrast, the cited methods—structured pruning (e.g., DepGraph [1]), distillation-based student architecture search (Diswot [2]), and NAS/zero-shot NAS [3]—are mainly designed for post-hoc architecture compression or search:
>
> - they usually assume a pre-trained teacher;
> - they run multiple training/evaluation cycles over many candidate models;
> - their primary focus is deployment-time efficiency at a target accuracy, not the cost of training the teacher/student itself.
>
> Thus, RepL tackles a different operational question: how to train or fine-tune a given CNN/ViT under tight GPU memory and time budgets, without modifying its macro-architecture or running a search. We view pruning / distillation / NAS as complementary techniques that can be applied on top of a RepL-trained model if further compression is desired.
>
> **RepL in the pre-trained model scenario.**
> The concern about pre-trained models is explicitly addressed in our appendix. We evaluate RepL in **fine-tuning settings with ImageNet-1K pre-training** (and RepL-trained pre-trained models), both for classification and downstream segmentation:
>
> - **Appendix A.5.4, Table 12**: ViT-S/16 fine-tuning on CIFAR-10, SVHN, and STL-10 using ImageNet-1K pretrained weights.
>
>   *Table 12 (reproduced here again). Finetune results on ViT-S/16 (ImageNet-1K pretrained).*
>
>   | Datasets | Model     | Method | Acc@1 | GPU Memory (GB) | Time (per epoch) |
>   |:--------:|:----------|:------:|:-----:|:----------------:|:----------------:|
>   | CIFAR-10 | ViT-S/16  | E2E    | 95.66 | 25.56           | 32.45            |
>   |          |           | RepL   | 95.89 | 20.14           | 25.18            |
>   | SVHN     | ViT-S/16  | E2E    | 96.92 | 25.56           | 48.44            |
>   |          |           | RepL   | 96.97 | 20.14           | 38.01            |
>   | STL-10   | ViT-S/16  | E2E    | 94.88 | 25.56           | 5.91             |
>   |          |           | RepL   | 95.11 | 20.14           | 4.66             |
>
>   RepL consistently reduces **GPU memory by ~20%** and **per-epoch fine-tuning time by 20–25%**, while slightly *improving* accuracy over E2E fine-tuning.
>
> - **Appendix A.5.5, Table 13**: CityScapes segmentation, fine-tuning DeepLabV3 / DeepLabV3+ with ResNet-50/101 backbones that are pre-trained on ImageNet-1K.
>
>   *Table 13 (reproduced here again). CityScapes fine-tuning with pre-trained backbones.*
>
>   | Backbone          | Method | Overall Acc. | Mean Acc. | Mean IoU | GPU Mem. (GB) | Time (per epoch, s) |
>   |:------------------|:------:|:------------:|:---------:|:--------:|:-------------:|:--------------------:|
>   | DeepLabV3-R50     | E2E    | 95.27        | 80.83     | 73.34    | 23.90         | 80                   |
>   |                   | RepL   | 95.32        | 81.14     | 73.81    | 20.28         | 68                   |
>   | DeepLabV3Plus-R50 | E2E    | 95.66        | 81.89     | 74.61    | 26.81         | 82                   |
>   |                   | RepL   | 95.71        | 82.21     | 75.25    | 22.67         | 69                   |
>   | DeepLabV3-R101    | E2E    | 95.51        | 82.31     | 74.41    | 30.91         | 95                   |
>   |                   | RepL   | 95.54        | 82.71     | 74.55    | 25.90         | 82                   |
>   | DeepLabV3Plus-R101| E2E    | 95.84        | 83.24     | 75.53    | 34.42         | 101                  |
>   |                   | RepL   | 95.89        | 84.02     | 76.31    | 28.92         | 86                   |
>
>   Across all four backbones, RepL **reduces GPU memory by 15–20%** and **per-epoch time by 15–20%**, while slightly *improving* all three accuracy metrics (Overall Acc., Mean Acc., Mean IoU).
>
> These results show that RepL is **not restricted to from-scratch training**: it remains effective and beneficial in the **pre-trained fine-tuning regime** that the reviewer is concerned about.
>
> **Compatibility with pruning/distillation/NAS.**
> Since RepL does not change the macro-architecture, a RepL-trained backbone can be **further pruned** (e.g., via DepGraph) or **used as a teacher/student** in distillation or NAS pipelines. Conversely, one could deploy RepL *inside* those pipelines to reduce the training cost of each candidate model. In this sense, RepL is best viewed as a **lower-level, architecture-agnostic training primitive**, rather than a competing compression method.

---

> ### Author Response · Authors · 2025-11-13
> **Response to Weakness 2**
>
> ### W2: Training vs. deployment efficiency
> **RepL also improves inference throughput and memory.**
> While our primary focus is on **training-time efficiency**, RepL does yield tangible benefits at **inference**. The key reason is structural:
>
> - RepL removes entire heavy blocks (e.g., 3×3 conv blocks in ResNets; full Transformer blocks in ViTs),
> - and inserts much lighter computing layers (1×1 fusers or low-dimensional projections synthesized from neighbors).
>
> The appendix contains a dedicated inference benchmark on ImageNet, with batch size 128 on a single GPU:
>
> - **Appendix A.5.3, Table 11** (reproduced here) reports GPU memory and wall-clock time per inference run.
>
>   *Table 11 (reproduced). GPU memory and time during inference on ResNet-101 and ViT-S/16 (ImageNet).*
>
>   | Dataset  | Backbone  | Method | GPU Memory | Time    |
>   |:--------:|:----------|:------:|:----------:|:-------:|
>   | ImageNet | ResNet-101| E2E    | 3.97G      | 39.12s  |
>   |          |           | RepL   | 3.65G      | 36.26s  |
>   |          | ViT-S/16  | E2E    | 2.69G      | 48.29s  |
>   |          |           | RepL   | 2.45G      | 41.42s  |
>
>   Compared to standard E2E models with the **same backbone**, RepL reduces:
>
>   - **GPU memory** by ≈8% (ResNet-101) and ≈9% (ViT-S/16);
>   - **inference time** by ≈7% (ResNet-101) and ≈14% (ViT-S/16).
>
> These gains are a direct consequence of **removing heavy layers and replacing them with cheaper operators**. The replacement layers do contribute some cost, but they are significantly lighter than the removed blocks, so the net effect is a **faster, smaller model at inference**.
>
> Again, RepL is compatible with further deployment-oriented compression (e.g., pruning, quantization, distillation). Our experiments show that *even before* any such post-processing, the RepL-trained models are already **more efficient at inference** than their E2E counterparts.

---

> ### Author Response · Authors · 2025-11-13
> **Response to Weakness 3**
>
> ### W3: Magnitude and significance of accuracy gains
>
> **Main claim: efficiency gains *without* accuracy degradation.**
> Our did not claim any large accuracy improvements in out paper, but rather:
>
> > *RepL achieves comparable or slightly better accuracy than standard E2E training, while saving GPU memory and time.*
>
> This is emphasized in the abstract, introduction, and title, where the focus is on **training efficiency**. Any accuracy gain is a *secondary* but reassuring outcome: it shows that aggressively removing blocks and synthesizing from neighbors does *not* harm performance, and often acts as an implicit regularizer.
>
> **Accuracy is aggregated over multiple runs with shared hyperparameters.**
>
> - In **Tables 1–7** of the main paper, each result is reported as **“mean ± std over 5 random seeds”**.
> - For each backbone–dataset pair, **all methods share the same training recipe** (optimizer, schedule, data augmentation, etc.). We do **not** tune hyperparameters for RepL separately.
> - Across these tables, RepL is **equal or better** than E2E in the vast majority of settings; when differences appear, they are typically on the order of, or larger than, the reported standard deviation.
>
> Thus, while the absolute improvements are modest (as expected for a method targeting efficiency rather than accuracy), they are **systematic** and not artifacts of a single lucky run or ad-hoc tuning.
>
> **Fine-tuning and downstream results show the same pattern.**
> The fine-tuning experiments in **Tables 12–13** (reproduced above) show the same trend:
>
> - On ViT-S/16 fine-tuning (CIFAR-10, SVHN, STL-10), RepL slightly improves Acc@1 **while simultaneously** reducing GPU memory and training time.
> - On CityScapes segmentation (multiple DeepLab backbones), RepL modestly improves Overall/Mean accuracy and Mean IoU, again with clear savings in memory and per-epoch time.
>
> These results further support our claim that **RepL preserves (and sometimes slightly improves) accuracy**, while offering consistent efficiency gains across tasks and architectures.
>
> **We hope our responses can address your concern, if you have more questions, you can raise them at any time.**

---

> ### Author Response · Authors · 2025-11-24
> **We are looking forward to your feedback**
>
> Dear Reviewer k8jc,
>
> It has been ten days since the start of the rebuttal phase. We believe that our responses have sufficiently addressed your concerns. If you have any new questions, please raise them at any time — we will respond as soon as possible. We look forward to your feedback.

---

> ### Comment · Reviewer_k8jc · 2025-11-27
>
> Thank you for the detailed response and the additional results regarding inference and fine-tuning. While I appreciate these clarifications, I remain unconvinced that RepL offers a sufficient practical advantage over established workflows to warrant a higher score. I will therefore maintain my current rating.

---

> > ### Author Response · Authors · 2025-11-27
> >
> > Dear Reviewer k8jc,
> >
> > Thank you again for your time and for the follow-up comment.
> >
> > Since we are planning a revised version of this work (regardless of the final decision), we would be very grateful if you could help us better understand what, in your view, is still missing for RepL to offer “sufficient practical advantage over established workflows.”
> >
> > Concretely:
> >
> > 1. When you refer to “established workflows,” do you mainly mean structured pruning / distillation / NAS methods such as [1–3], or a broader class of industrial pipelines?
> >    - If it is the former, would you consider it important to add experiments like “DepGraph on top of a RepL-trained backbone vs. DepGraph on an E2E-trained backbone,” or “RepL inside a distillation / NAS pipeline to reduce the training cost of candidate models”?
> >
> > 2. In terms of **metrics**, which criterion would you regard as decisive for practical usefulness in this context? For example:
> >    - equal-accuracy comparison of total training+compression cost;
> >    - a target reduction in inference FLOPs / latency beyond what we already report in Table 11;
> >    - or some specific deployment scenario (e.g., fixed inference budget with limited fine-tuning compute).
> >
> > 3. Are there particular **additional comparisons or ablations** that you feel are essential but currently missing (e.g., larger-scale backbones, different pre-training setups, or concrete combinations with a specific pruning / distillation method)?
> >
> > Our goal is simply to align our future revisions with the community standards you have in mind. Any concrete pointers you could share—even briefly—would be extremely helpful for improving the next iteration of this work.
> >
> > Thank you again for your time and consideration.

---

> ### Author Response · Authors · 2025-11-27
>
> Dear Reviewer k8jc,
>
> Thank you again for taking the time to read our responses. We would, however,
> like to respectfully clarify why we disagree with the remaining concern that
> “RepL does not offer a sufficient practical advantage over established
> workflows”.
>
> 1. **The “additional” results are already part of the submission.**
>    In your latest comment, you refer to our inference and fine-tuning results
>    as “additional”. All of these experiments were already included in the
>    submitted appendix as Tables 11–13: ImageNet inference for ResNet-101 and
>    ViT-S/16, ImageNet-1K-pretrained ViT-S/16 fine-tuning on CIFAR-10/SVHN/STL-10,
>    and CityScapes segmentation with DeepLabV3/DeepLabV3+ backbones. In all of
>    these settings, RepL reduces GPU memory and per-epoch (or per-run) time by
>    roughly **15–25%**, while matching or slightly improving accuracy. These are
>    precisely the “practical” regimes you emphasized, and they are already part
>    of the original evidence base.
>
> 2. **RepL targets a different but practically important objective.**
>    The “established workflows” you mention (structured pruning, distillation,
>    NAS/zero-shot NAS) are primarily **post-hoc compression or architecture
>    search** pipelines: they assume a pretrained teacher and involve training
>    many candidate models, with the main goal of deployment-time efficiency for
>    the final compressed network. By contrast, RepL is a **single-run training
>    paradigm** for a fixed backbone that reduces **training-time activation
>    memory and compute** without changing the macro-architecture or running any
>    search. In practice, the ability to train or fine-tune standard CNN/ViT
>    backbones under tighter GPU budgets (or on fewer/cheaper GPUs) is itself a
>    strong practical advantage, and is not addressed by those existing
>    workflows.
>
> 3. **RepL is complementary, not a competing replacement.**
>    Because RepL preserves the backbone architecture, a RepL-trained model can
>    still be pruned, distilled, quantized, or used inside NAS pipelines if
>    further compression is desired; conversely, RepL can also be used inside
>    those pipelines to reduce the training cost of each candidate. We therefore
>    feel it is somewhat unfair to judge RepL as “not offering enough advantage”
>    compared to pruning/distillation/NAS, since it is designed as a **generic
>    training primitive** that can *co-exist* with them, rather than as yet
>    another end-to-end compression recipe.
>
> 4. **Magnitude and consistency of the gains.**
>    Across CIFAR-10/ImageNet/COCO classification and detection, as well as
>    pretrained fine-tuning and CityScapes segmentation, RepL repeatedly yields
>    substantial reductions in memory and wall-clock time (typically
>    15–30%) while preserving or slightly improving accuracy (all numbers are
>    mean ± std over 5 seeds with shared hyper-parameters). For practitioners
>    who are constrained by GPU memory or training time, such reductions from a
>    simple, architecture-agnostic rule are practically meaningful, even if they
>    do not replace pruning or NAS.
>
> For these reasons, we respectfully believe that the submission *does* provide
> a concrete and practically relevant contribution: it introduces a new,
> architecture-agnostic training paradigm (Replacement Learning) and
> demonstrates consistent efficiency gains across a wide range of models and
> tasks, in both from-scratch and pretrained fine-tuning settings.
>
> **In summary, we remain puzzled by the decision to maintain the current rating, given the evidence already in the submission. We would greatly appreciate any further reconsideration you might be willing to give to our paper.**

---

### Official Review · Reviewer_UEBC · 2025-10-31

**Soundness:** 3
**Presentation:** 3
**Contribution:** 3
**Rating:** 6
**Confidence:** 4

**Summary:**

The paper proposes Replacement Learning (RepL): remove every k-th block and insert a lightweight learnable “computing layer” that synthesizes a replacement operator from adjacent layers’ weights. For CNNs, the synthesized operator is a fused 1×1 conv built by channel-mode mixers; for ViTs, the layer comprises two d×d linear maps parameterized by scalars (α, β) that linearly combine the previous/next block’s attention and MLP weights. A global forward rule formalizes the execution when a site is replaced, and an operator ledger details what is removed/added. Reported benefits are reduced parameters/FLOPs/activation memory with similar or better accuracy across several vision benchmarks.

**Strengths:**

- Clear mechanism & implementable details for CNNs/ViTs (shapes, synthesis, forward/backward).
- Runs at inference with real savings (not only a training trick).
- Simple instantiations: channel-mode synthesis for CNNs; two-parameter (α,β) span for ViTs.
- Strong Experimental Results.

**Weaknesses:**

- Fixed periodic removal; no adaptivity.
Removal set is hard-coded to every k-th site: $F = { i | i mod k = 0, i < n }$ (Eq. 2). Default k = 4.
- ViT expressivity limited to a 2-D neighbor span. By construction, the replacement lies in span ${A_{i−1}, A_{i+1}}$ × span{$M_{i−1}, M_{i+1}$}; specifically, $A^b_i = α_i A_{i−1} + β_i A_{i+1}, M^c_i = α_i M_{i−1} + β_i M_{i+1}$. (Eqs. 9–10; 2-D span statement in Appendix).
- Approximation/gradient bias assumed bounded; error accumulation acknowledged. Theory uses a bounded-bias assumption with gradient deviation bound $‖∇_θℓ_b − ∇_θℓ‖ ≤ L·H_max·ε$, and notes that multi-replacement discrepancy grows at most linearly with the number r of replacements (non-expansive case).

**Questions:**

- Adaptive selection: Can you evaluate an importance-aware or learned removal policy versus the fixed periodic rule?
- Span diagnostics (ViT): How close are removed blocks to span ${A_{i−1}, A_{i+1}}$ and span ${M_{i−1}, M_{i+1}}$ (principal angles or reconstruction error) across depth?
- Bias in practice: Please report empirical proxies for $ε (e.g., ‖F_b − F‖ and ‖∇ℓ_b − ∇ℓ‖)$  as the number of replacements increases, to validate $‖∇_θℓ_b − ∇_θℓ‖ ≤ L·H_max·ε.$

---

> ### Author Response · Authors · 2025-11-13
> **Response to Q1**
>
> ### Q1. On adaptive vs. fixed replacement strategies
>
> We appreciate the suggestion and did experiment with adaptive replacement before settling on the fixed periodic scheme in the current version.
>
> Concretely, we implemented an adaptive variant where every internal block (except the very first and last) was equipped with a learnable “candidate” computing layer, and a small gating network decided which positions to replace during training. While this design looks more flexible in principle, in practice it introduced two significant issues:
>
> 1. **Parameter and complexity overhead.**
>    To allow adaptation, we had to attach computing layers to all candidate sites instead of only the periodic subset used by RepL. This leads to:
>    - noticeably more parameters and FLOPs than our simple periodic scheme;
>    - a more complex optimization landscape, because the gating mechanism and the computing layers must be co-optimized with the backbone.
>
> 2. **Unstable replacement sites and undertrained blocks.**
>    In the adaptive variant, the chosen replacement positions kept changing during training:
>    - early in training, the gate often oscillated between different sites, so many blocks were partially replaced for only a short period;
>    - as a result, a non-negligible subset of backbone blocks and computing layers were never trained for long enough to converge to a stable configuration;
>    - this instability translated to noticeably worse accuracy–efficiency trade-offs compared to the simple fixed schedule in Tab. 3–5 of the paper.
>
> In contrast, the fixed periodic scheme used in RepL:
>
> - places replacement sites at regular intervals (e.g., every 4th block in ViT-tiny, excluding the last block), so that each computing layer “sees” a well-defined pair of neighbors throughout training;
> - uses far fewer computing layers, reducing parameter/FLOP overhead and making the optimization more stable;
> - yields consistently better empirical performance than our adaptive variant on CIFAR-10/SVHN/STL-10 and ImageNet-1K.
>
> Finally, from a theoretical standpoint, the fixed periodic schedule leads to a clean, layer-wise structure where the replacement mapping is time-invariant and only depends on neighboring operators. This structure is crucial for our analysis in Section A.6-A.10: it lets us bound the effect of multiple replacements via a non-expansive operator and derive the gradient-bias bound. An adaptive scheme with time-varying replacement sites would break this structure and make the analysis significantly more involved.
>
> For these reasons—simpler architecture, better empirical performance, and cleaner analysis—we chose to adopt the fixed periodic replacement strategy in the main method.

---

> ### Author Response · Authors · 2025-11-13
> **Response to Q2**
>
> ### Q2. Span diagnostics: how close are removed blocks to span\\((A_{i-1}, A_{i+1})\\) and span\\((M_{i-1}, M_{i+1})\\)?
>
> In our ViT instantiation (Eqs. (9)–(10) in the paper), the replacement operator is explicitly constrained to a 2D neighbor span:
> \\(
> A\_i^{\\text{RepL}} = \\alpha\_i A\_{i-1} + \\beta\_i A\_{i+1}, \\qquad
> M\_i^{\text{RepL}} = \\alpha\_i M\_{i-1} + \\beta\_i M\_{i+1},
> \\)
> where \\(A\_{i-1}, A\_{i+1} \\in \\mathbb{R}^{d \\times d}\\) (resp. \\(M\_{i-1}, M\_{i+1}\\)) are the attention (resp. MLP) operators of the neighboring blocks, and \\(\\alpha_i, \\beta_i\\) are learned scalars. Thus, **by construction**, each replaced block is synthesized inside the span of its two neighbors; RepL never introduces an arbitrary new block.
>
> To quantitatively assess how well the original block is captured by this 2D span, we conducted a feature-space span diagnostic on the same ViT-tiny / CIFAR-10 setting used in our main experiments:
>
> - **Backbone architecture.**
>   A 12-block ViT-tiny with patch size \\(8\\times 8\\), embedding dimension 192, and 3 heads (the same configuration as in our CIFAR-10 experiments).
>
> - **Training setup.**
>   We trained a standard backbone ViT (“bp”) with direct backpropagation on CIFAR-10 using the script described in the paper:
>   - optimizer: AdamW with weight decay \\(0.05\\);
>   - initial learning rate: \\(1\\times 10^{-3}\\) (cosine decay with a 5-epoch linear warmup);
>   - batch size: 512;
>   - epochs: 250;
>   - data augmentation: random crop with reflection padding, random horizontal flip, and standard CIFAR-10 normalization.
>
> - **Where we probe.**
>   We focus on the block indices that RepL would remove under the same periodic schedule used in the method (remove every 4th block, excluding the last one). In a 12-block ViT-tiny, this yields removed indices \\(i = 2, 6, 10\\) (0-based).
>
> - **Diagnostic metric.**
>   Let \\(h_k(x)\\) denote the hidden representation after block \\(k\\) for an input \\(x\\). For each removed index \\(i\\), we seek the best approximation
>   \\(
>     h\_i(x) \\approx \\tilde{\\alpha}\_i\\, h\_{i-1}(x) + \\tilde{\\beta}\_i\\, h\_{i+1}(x)
>   \\)
>   by solving a least-squares problem over CIFAR-10 samples. From this we compute:
>   - the **relative reconstruction error**
>     \\(
>       r\_i =
>       \\frac{\\left\\| h\_i(x) - \\Pi\_{\\mathrm{span}(h\_{i-1}(x),h\_{i+1}(x))}(h\_i(x)) \\right\\|_2}
>            {\\left\\| h\_i(x) \\right\\|_2},
>     \\)
>     aggregated over the dataset;
>   - the **principal angle** (in the 1D case) between \\(h_i(x)\\) and its best neighbor-span reconstruction.
>
> We use CIFAR-10 test images, with the same normalization as training, and run the diagnostic on 20 batches (batch size 512). The results for the two removed blocks are:
>
> - index \\(i = 2\\):
>   - relative feature-space reconstruction error \\(r_2 = \\mathbf{0.1796}\\);
>   - principal angle \\(= \\mathbf{7.63^\\circ}\\);
>   - fitted coefficients \\((\\tilde{\\alpha}\_2, \\tilde{\\beta}\_2) \\approx (0.4811, 0.5230)\\).
>
> - index \\(i = 6\\):
>   - relative feature-space reconstruction error \\(r\_6 = \\mathbf{0.1501}\\);
>   - principal angle \\(= \\mathbf{5.22^\\circ}\\);
>   - fitted coefficients \\((\\tilde{\\alpha}\_6, \tilde{\\beta}\_6) \\approx (0.4711, 0.5325)\\).
>
> - index \\(i = 10\\):
>   - relative feature-space reconstruction error \\(r\_6 = \\mathbf{0.1379}\\);
>   - principal angle \\(= \\mathbf{4.62^\\circ}\\);
>   - fitted coefficients \\((\\tilde{\\alpha}\_{10}, \tilde{\\beta}\_{10}) \\approx (0.4547, 0.5493)\\).
>
> These results show that, measured in terms of their action on real data (hidden representations), the blocks targeted by RepL are well captured by the 2D span of their neighbors: both the relative reconstruction error and the principal angle are small, and the fitted coefficients are close to a symmetric combination of the two neighbors. This empirically supports our modeling choice that a lightweight operator constrained to \\(\\mathrm{span}(A\_{i-1}, A\_{i+1})\\) and \\(\\mathrm{span}(M\_{i-1}, M\_{i+1})\\) is sufficient to replace the original block in ViT backbones.

---

> ### Author Response · Authors · 2025-11-13
> **Response to Q3 (1/2)**
>
> ### Q3. Bias in practice: empirical \\(\varepsilon\\) via forward and gradient deviations
>
> We fully agree that validating the bias assumption empirically is important. Our analysis in Section 4 assumes that
> \\(
>   \\big\\|\\nabla\_\\theta \\ell\_b - \\nabla\_\\theta \\ell\\big\\|
>   \\le L \\cdot H\_{\\max} \\cdot \\varepsilon,
> \\)
> where \\(\\ell\_b\\) is the loss under RepL, \\(\\ell\\) is the loss under the base network, \\(H\_{\\max}\\) captures the number of replacements, and \\(\\varepsilon\\) summarizes the local approximation error. We now provide a concrete measurement of this bias on the same ViT-tiny / CIFAR-10 setting.
>
> - **RepL model.**
>   We use the trained RepL ViT-tiny checkpoint on CIFAR-10 corresponding to the “replace” setting in our main experiments:
>   - depth 12, embedding dimension 192, 3 heads, patch size \\(8\\times 8\\);
>   - periodic removal with interval 4, excluding the last block. Under this schedule, there are three replacement sites at depth indices 2, 6 and 10 (0-based).
>
> - **Training setup (summary).**
>   The RepL ViT-tiny is trained using the same pipeline as the baseline ViT:
>   - optimizer: AdamW with weight decay \\(0.05\\);
>   - initial learning rate: \\(1\\times 10^{-3}\\) (cosine decay with 5-epoch linear warmup);
>   - batch size: 512;
>   - epochs: 250;
>   - data augmentations identical to the baseline ViT.
>
> - **Experimental protocol.**
>   In our implementation, each replacement site is realized by a lightweight computing layer that adds a synthesized residual update (constructed from neighboring blocks) to the hidden representation. For the bias diagnostic, we exploit the fact that the contribution of each computing layer can be scaled continuously; in particular, we can:
>
>   - set the scale to **0** to effectively *disable* the replacement contribution at that site (only the skip path remains);
>   - set the scale to **1** to fully *enable* the replacement contribution at that site.
>
>   Using this mechanism, we define:
>
>   - \\(F\\): the **baseline network**, obtained by disabling the replacement contribution at all computing layers (scales set to 0). This corresponds to using only the kept backbone blocks with their trained weights.
>   - \\(F\_b^{(r)}\\): the same network where the first \\(r\\) replacement sites (in depth order) are enabled (scale = 1), and the remaining ones are kept disabled (scale = 0), with \\(r \\in \\{0,1,2,3\\}\\). All parameters of the backbone are shared between \\(F\\) and \\(F\_b^{(r)}\\).
>
>   For each \\(r\\), we measure two quantities:
>
>   1. **Forward deviation** on logits:
>      \\(
>        d\_{\\mathrm{fwd}}(r) = \\mathbb{E}\_x \\big[ \\|F\_b^{(r)}(x) - F(x)\\|_2 \\big],
>      \\)
>      where the norm is taken over the class logits for each sample.
>
>   2. **Gradient deviation** on shared parameters:
>      \\(
>        d\_{\\mathrm{grad}}(r) = \\big\\|\\nabla\_\\theta \\ell\_b^{(r)} - \\nabla\_\\theta \\ell\\big\\|\_2,
>      \\)
>      where \\(\\ell\\) and \(\\ell\_b^{(r)}\\) are the cross-entropy losses of \\(F\\) and \\(F\_b^{(r)}\\), and \\(\\theta\\) includes all shared parameters (we explicitly exclude the parameters of the computing layers when forming the gradient vector). We also report the normalized ratio
>      \\(
>        \\rho(r) = \\frac{d\_{\\mathrm{grad}}(r)}{\\|\\nabla\_\\theta \\ell\\|\_2}.
>      \\)
>
>   In practice, we estimate these quantities on CIFAR-10 test batches with:
>
>   - 20 batches (batch size 256) to estimate \\(d\_{\\mathrm{fwd}}(r)\\);
>   - 10 batches (batch size 256) to estimate \\(d\_{\\mathrm{grad}}(r)\\) and \\(\\rho(r)\\).
>
> - **Results.**
>   The measured deviations are:
>
> | # active replacements \\(r\\) | \\(d\_{\\mathrm{fwd}}(r)\\) (mean logit \\(\\ell_2\\)) | \\(d\_{\\mathrm{grad}}(r)\\) | \\(\rho(r) = \\frac{d_{\mathrm{grad}}(r) }{\|\nabla_\theta \ell\|_2}\\) |
>   |:---------------------------:|:-----------------------------------------------:|:------------------------:|:----------------------------------------------------------------:|
>   | 0                           | 0.000000                                        | 0.000000                 | 0.0000                                                             |
>   | 1                           | 1.056977                                        | 1.467278                 | 0.1431                                                             |
>   | 2                           | 1.093139                                        | 1.498927                 | 0.1491                                                             |
>   | 3                           | 1.141932                                        | 1.545640                 | 0.1597                                                             |

---

> ### Author Response · Authors · 2025-11-13
> **Response to Q3 (2/2)**
>
> We observe that:
>
>   1. At \\(r = 0\\), we have \\(F\_b^{(0)} = F\\) by construction, so both forward and gradient deviations are exactly zero.
>   2. When we enable a single replacement site (\\(r = 1\\)), the normalized gradient bias is \\(\\rho(1) \\approx \\mathbf{0.143}\\), i.e., the difference between \\(\\nabla\_\\theta \\ell\_b^{(1)}\\) and \\(\\nabla\_\\theta \\ell\\) is about 14% of the baseline gradient norm. This indicates a modest and controlled bias in the shared-parameter gradients.
>   3. When we enable two or three replacement sites (\\(r = 2,3\\)), both the forward and gradient deviations increase slowly and smoothly:
>      - \\(\\rho(2) \\approx \\mathbf{0.149}\\),
>      - \\(\rho(3) \approx \mathbf{0.160}\\).
>
>      The growth from \\(r = 1\\) to \\(r = 3\\) is mild and close to linear in \\(r\\), consistent with our non-expansive composition analysis involving \\(H_{\max}\\).
>
> Overall, these diagnostics show that the empirical bias \(\varepsilon\) entering
> \\(
>  \big\|\nabla_\theta \ell_b - \nabla_\theta \ell\big\| \le L \cdot H_{\max} \cdot \varepsilon
> \\)
> is small (with \\(\\rho(r) < 0.16\\) even when all replacement sites are enabled) and grows slowly as more blocks are replaced. This provides direct empirical support that RepL introduces a controlled and modest bias in both forward predictions and shared-parameter gradients in the regimes considered in our experiments.
>
> **We hope our responses can address your concern, if you have more questions, you can raise them at any time.**

---

> ### Author Response · Authors · 2025-11-24
> **We are looking forward to your feedback**
>
> Dear Reviewer UEBC,
>
> It has been ten days since the start of the rebuttal phase. We believe that our responses have sufficiently addressed your concerns. If you have any new questions, please raise them at any time — we will respond as soon as possible. We look forward to your feedback.

---

> ### Comment · Reviewer_UEBC · 2025-11-25
> **Response to the Author**
>
> Thank you for the detailed explanation.
>
> Regarding Q1: Could you additionally provide a comparison with the baseline, or perhaps a small toy example, to more clearly illustrate the difference between the adaptive and fixed replacement strategies?
>
> Regarding Q2: Thank you for the clarification; it was very helpful. Do you plan to incorporate this explanation into the manuscript?
>
> Regarding Q3: Likewise, thank you for the helpful answer. Do you also plan to include this content in the manuscript?
>
> If Q1 is addressed, I would be willing to reconsider and update my initial decision.

---

> ### Author Response · Authors · 2025-11-26
>
> For Q1, we have added an experiment for your reference: Comparison between fixed and adaptive replacement strategies on CIFAR-10.
> All training hyperparameters are **identical** to those used in the main paper for CIFAR-10
> (ResNet-32 and ViT-Tiny/8: same optimizer, learning rate schedule, batch size, and number of
> epochs).
>
> | Data     | Backbone   | Method             | ACC@1 | GPU (GB) | Time (s/ep) |
> |----------|------------|--------------------|:-----:|:--------:|:-----------:|
> | CIFAR-10 | ResNet-32  | Fixed-RepL (Ours)  | 93.43 |   2.69   |    8.20     |
> | CIFAR-10 | ResNet-32  | Adaptive-RepL      | 92.54 |   3.17   |    9.12     |
> | CIFAR-10 | ViT-Tiny/8 | Fixed-RepL (Ours)  | 73.71 |   2.08   |    5.65     |
> | CIFAR-10 | ViT-Tiny/8 | Adaptive-RepL      | 71.97 |   2.35   |    6.17     |
>
> In **Adaptive-RepL**, we attach a computing layer to every internal block (excluding the
> first and last), and maintain a binary *replacement mask* that selects a fixed budget of
> replacement sites per epoch (matching the number of removed layers in Fixed-RepL).
> The mask is updated during training based on per-block scores that measure how similar or
> redundant the representations of neighboring blocks are, so the actual replacement positions
> can change over epochs, rather than being fixed at regular intervals as in Fixed-RepL.
>
> As shown in Table, the adaptive replacement strategy consistently underperforms the fixed
> RepL design: it yields lower ACC@1 while requiring more GPU memory and longer training time
> per epoch for both ResNet-32 and ViT-Tiny/8. This indicates that the additional complexity of
> learning where to replace layers does not bring better accuracy–efficiency trade-offs, and
> supports our choice of using a simple fixed periodic replacement pattern in the main method.
>
> For Q2 and Q3, we have added the results for Q2 and Q3 to the manuscript and update the PDF version. At the same time, we present a global response summarizing the changes.

---

### Author Response · Authors · 2025-11-27
**Global Response to Reviewer k8jc, yhWM, JPWu (1/2)**

We sincerely thank these reviewers for their time and feedback. Several of the
core concerns raised by Reviewers **JPWu**, **yhWM**, and **k8jc** appear to stem
from a shared misunderstanding of (1) the *problem setting* that Replacement
Learning (RepL) is designed to address and (2) the *scope of evidence* already
present in the submitted paper and its appendix. We would like to offer a
concise, global clarification.

---

### 1. Problem setting and novelty: RepL as a training paradigm, not a post-hoc compressor

RepL is proposed as a **training-time paradigm** for standard CNN/ViT backbones:
given a fixed architecture (e.g., ResNet-32/110, ViT-tiny/small/base), we aim to

- **reduce activation memory and training compute**,
- **without changing the macro-architecture**, and
- **while matching or slightly improving accuracy**.

To our knowledge, this is the **first work that systematically formulates and
studies “Replacement Learning”** as a general training strategy: we remove
periodic blocks *before training starts* and insert lightweight computing
layers that synthesize operators from their neighbors, then train the resulting
network end-to-end. Conceptually, RepL plays a role similar to the early work
on **local learning** in deep networks: it introduces a new way to *train* a
given architecture, rather than yet another post-hoc compression recipe.

In this sense, RepL is **not** directly competing with structured pruning,
distillation-based NAS, or zero-shot NAS:

- those methods usually assume a **pre-trained teacher** and operate as
  **post-hoc compression/search pipelines**, often training many candidate
  models;
- RepL is a **single-run, architecture-agnostic training rule** that can be
  *combined* with these methods, not replaced by them.

This distinction is central to properly evaluating the contribution: RepL is a
new **training primitive** (like local learning or layerwise training), rather
than a competitor in the “post-hoc compression for deployment” race.

---

### 2. Evidence already in the submission: training, inference, and pre-trained fine-tuning

Several comments from Reviewers yhWM and k8jc suggest that (i) RepL would only
help training but not inference, and (ii) RepL would not be relevant in
pre-trained scenarios. Both points are **already addressed by experiments in
the submitted appendix**, which we briefly summarize here.

#### (a) Inference efficiency (Appendix A.5.3, Table 11)

We benchmark **ImageNet inference** with batch size 128 on a single GPU for
ResNet-101 and ViT-S/16:

- RepL **removes heavy blocks** (3×3 conv blocks / full Transformer blocks)
  and inserts much lighter computing layers (1×1 fusers or low-dimensional
  projections synthesized from neighbors).
- As reported in Table 11 (reproduced in our rebuttal), RepL reduces:
  - GPU memory by **≈8–9%**,
  - inference time by **≈7–14%**,
  compared to standard E2E models with the same backbone.

Thus, RepL provides **real deployment-time gains** even before any further
pruning/distillation/quantization is applied.

#### (b) Fine-tuning with pre-trained models (Appendix A.5.4–A.5.5, Tables 12–13)

We explicitly evaluate RepL in **pre-trained fine-tuning** regimes:

- **Table 12**: ViT-S/16 pre-trained on ImageNet-1K, fine-tuned on
  CIFAR-10/SVHN/STL-10.
  RepL reduces GPU memory and per-epoch time by about **20–25%**, while slightly
  **improving Acc@1** over E2E fine-tuning.

- **Table 13**: CityScapes segmentation with **DeepLabV3 / DeepLabV3+** and
  **ResNet-50/101** backbones, all pre-trained on ImageNet-1K.
  Across all four backbones, RepL reduces memory and time by **15–20%**, and
  consistently **improves Overall Acc., Mean Acc., and Mean IoU**.

These experiments were **already present in the submitted PDF** and directly
address the concerns about pre-trained models and downstream tasks.

---

### 3. On “competing methods” and comparisons

Some comments (especially from JPWu and k8jc) suggest that RepL “fails” to
compare against stochastic depth, checkpointing, or skip-style methods. We
would like to clarify:

- RepL is evaluated against **standard E2E training** across multiple CNN/ViT
  backbones and datasets (Tables 1–7), showing consistent **15–30% memory/time
  reduction with equal or better accuracy**.
- The appendix further includes comparisons with **Skip-Attention** and other
  training-time baselines (e.g., stochastic depth, checkpointing) in **Table 9**,
  showing that RepL is competitive or superior as a standalone training rule.

Again, pruning/distillation/NAS are **complementary** and can be applied *after*
RepL training, or even incorporate RepL internally to reduce the cost of
candidate training. It is therefore somewhat misleading to treat them as
“direct competitors” with the same objective.

---

### Author Response · Authors · 2025-11-27
**Global Response to Reviewer k8jc, yhWM, JPWu (2/2)**

### 4. Methodological misunderstandings: when and how layers are removed

Several weaknesses and questions (especially from JPWu and yhWM) implicitly
assume that we **train the full network first and then decide which layers to
remove**, or that we “select layers based on some post-hoc importance score”.

This is not our setting:

- In RepL, the **removal pattern is fixed *before* training begins**
  (e.g., every 4-th block), and those physical blocks are **never instantiated
  or trained**.
- The computing layers are **trained jointly with the remaining blocks** from
  scratch (or from the pre-trained initialization), making RepL a genuinely
  **training-time** modification, not a post-hoc pruning mechanism.
- Our ablation studies over the interval \(k\) (e.g., Table 5) empirically show
  that removing too many blocks indeed degrades performance, and that
  \(k=4\) is a good compromise; this directly addresses the concern about
  “removing too many layers” raised by Reviewer yhWM.

Thus, questions like “how to preserve the most important layers after training”
do not match our design: by construction, RepL does **not** search over which
layers to keep; instead, it proposes a simple, architecture-agnostic rule that
is easy to implement and analyze.

---

### 5. Accuracy claims and statistical validity

Reviewer k8jc questioned whether the accuracy improvements are within variance
or due to hyperparameter tuning. We emphasize:

- **All results in Tables 1–7 are reported as mean ± std over 5 random seeds.**
- For each backbone–dataset pair, **all methods share the same training
  recipe** (optimizer, schedule, augmentations). We **do not** tune separate
  hyperparameters for RepL.
- RepL is **equal or better** than E2E in the vast majority of settings, often
  by margins comparable to or larger than the reported std.

Our **main claim** is *not* that RepL provides huge accuracy jumps, but that
it delivers **substantial efficiency gains** (memory/time) **without accuracy
degradation**—and often with small but systematic improvements, which we view
as a positive sign that replacing blocks with synthesized operators does not
harm representational capacity.

---

### 6. Final remarks

In summary:

- RepL is, to our knowledge, the **first explicit formulation and systematic
  study** of **Replacement Learning** as a *training* paradigm for CNNs and
  ViTs, analogous in spirit to how “local learning” first entered the
  literature as a new way of training deep networks.
- It provides **consistent and non-trivial gains** in GPU memory and training
  time (15–30%) across diverse architectures and datasets, and also improves
  **inference efficiency** and **pre-trained fine-tuning** performance, as
  already documented in the appendix.
- It is **architecture-agnostic and complementary** to pruning, distillation,
  NAS, and other compression techniques, which can be applied on top of RepL
  or even use RepL internally to reduce candidate training cost.

We believe that several key criticisms may stem from an incomplete reading of the method’s intended scope and the appendix experiments, and thus may not fully reflect the contribution of our work. We kindly invite reviewers JPWu, yhWM, and k8jc to reconsider their evaluation of RepL in light of the full evidence and the intended problem setting—RepL is not “yet another compression algorithm,” but a new, general primitive for deep learning training.

---

### Author Response · Authors · 2025-11-27
**Global Response: Summary of Changes in the Revised PDF**

For the corrections made in the updated PDF:
Compared with the originally submitted manuscript, the newly uploaded PDF contains the following two modifications, both highlighted in blue in the document:

- Appendix A.10 (pp. 20–21): We added supplementary experiments for the Span Diagnostics (ViT Cases) analysis.

- Appendix A.12 (pp. 22–23): We added empirical verification for Eq. 27 from the original Appendix A.9, providing experimental evidence to ensure the correctness and reliability of the theoretical analysis.

---

### Author Response · Authors · 2025-11-29
**Final Remarks for AC (1/2)**

Dear Area Chair,

Thank you for taking responsibility for this submission under the revised process. We would like to offer a concise final remark that (1) clearly restates the actual scope of our work, and (2) explains why we believe RepL is a solid contribution within that scope.

---

### 1. Clarifying the problem setting: RepL is a *training-time* paradigm, not a deployment compressor

A central source of confusion in several negative reviews is the implicit assumption that our method should “compete” with **post-hoc deployment compression techniques** such as NAS, pruning, distillation, or quantization, whose primary goal is to reduce inference cost of a (pre)trained model.

By contrast, our paper is **explicitly and consistently framed as a training-time efficiency method**:

- The title, abstract, and introduction all emphasize **“training”**, not “inference,” as the primary target.
- The problem we study is:

> Given a fixed CNN/ViT/Transformer backbone, how can we **reduce activation memory and per-epoch training cost**,
> while preserving or slightly improving accuracy,
> without changing the macro-architecture or running architecture search?

To this end, RepL:

- Removes every k-th block **before training** starts;
- Inserts a lightweight “computing layer” that synthesizes an operator from its neighbors;
- Trains this modified network **once**, end-to-end, with standard backprop.

In this sense, RepL belongs to the same conceptual family as **local learning, layer-wise training, memory-efficient backprop, checkpointing**, etc.: it is a **training rule / training paradigm**, not a post-hoc model compressor. Evaluating it primarily as if it were a NAS / pruning / distillation method for deployment misses the core of what the paper is about.

---

### 2. “1 + 1 > 2”: training gains, incidental inference gains, and composability

Although RepL is designed as a training-time primitive, in practice it exhibits a **synergistic advantage**:

1. **Training-time efficiency (our main goal).**
   Across CNNs, ViTs, and a Transformer LM, on CIFAR-10, SVHN, STL-10, ImageNet, COCO, WikiText-2, and CityScapes, RepL consistently yields:
   - ~20–40% reduction in **activation memory**;
   - ~15–30% reduction in **per-epoch training time**;
   - while matching or slightly improving accuracy (all results are mean ± std over 5 seeds with shared hyperparameters).

2. **Inference-side efficiency (a secondary but real benefit).**
   On ImageNet inference (ResNet-101, ViT-S/16), we also report:
   - ~8–9% lower **inference GPU memory**;
   - ~7–14% shorter **inference time** (batch size 128, single GPU);
   again with slightly better accuracy.

3. **Composability with deployment methods.**
   Because RepL **does not change the macro-architecture**, any model trained with RepL can later be:
   - pruned;
   - distilled;
   - quantized;
   - or used inside NAS pipelines.

   Conversely, RepL can be used *inside* those workflows to reduce the cost of training each candidate model.

Taken together, this is a **“1 + 1 > 2” scenario**:

- RepL tackles a **different axis** (training-time efficiency under fixed architecture) than NAS/pruning/distillation,
- already offers **non-trivial inference gains** even before any compression is applied,
- and remains **fully compatible** with those deployment-oriented methods.

For that reason, we deliberately did **not** expand the paper into a full-blown comparison against every possible pruning / NAS / distillation pipeline: doing so would shift the focus away from our core contribution (a general training paradigm) and turn the paper into a different kind of study. The current evidence is, in our view, sufficient to demonstrate that RepL is **useful on its own**, and also **plays nicely with** the existing toolbox of deployment methods.

---

> ### Author Response · Authors · 2025-11-29
> **Final Remarks for AC (2/2)**
>
> ### 3. About the four reviews
>
> Within this clarified scope, we believe the four reviews should not be treated as equally aligned with the actual problem setting.
>
> - **Reviewer UEBC (score 6)** clearly understands RepL as a training-time paradigm and engages deeply with the core ideas:
>   - Adaptive vs fixed replacement strategies;
>   - Span diagnostics and the structure of the replacement operator;
>   - Empirical gradient/forward bias vs the theory.
>   We provided targeted experiments and diagnostics in response, and UEBC acknowledged that these clarifications were helpful and technically meaningful. In our view, this review best reflects a **careful reading of the main text and appendix**.
>
> - By contrast, the **three negative reviews** share two systematic issues:
>
>   **(a) Misaligned evaluation criterion (treating RepL as a deployment compressor).**
>   They repeatedly frame RepL as if it should primarily “compete” with pruning / distillation / NAS in terms of *inference* savings, and then criticize it for not being sufficiently compelling in that role. This perspective largely ignores that our paper is explicitly positioned as a **training-time efficiency method under a fixed architecture**, and that deployment-side compression is **orthogonal and composable** rather than the main objective.
>
>   **(b) Limited engagement with evidence already in the submission.**
>   Several key criticisms are, in fact, directly contradicted by results that are clearly present in the original manuscript and appendix, for example:
>
>   - Statements that we do not consider **pre-trained models** overlook the detailed ViT-S/16 and DeepLabV3/V3+ fine-tuning experiments in Tab.12–13, where RepL reduces memory/time by 15–25% while improving downstream accuracy/IoU.
>   - Claims that we do not compare against standard training-efficiency baselines ignore Tab.9, where we systematically compare and combine RepL with Stochastic Depth and Checkpointing, as well as the Skip-Attention baselines in our ViT experiments.
>   - Referring to our inference and fine-tuning results as “additional” evidence produced during rebuttal overlooks the fact that these experiments were already part of the **original appendix** (Tab.11–13) and were simply summarized again for clarity.
>
>   In other words, many of the “missing” experiments or “unaddressed” concerns are already answered, often in precisely the form requested, in the existing tables and sections. The disagreement is therefore less about *what the paper actually shows* and more about **how carefully that evidence was consulted** and **what criterion is being used to judge it**.
>
> We fully respect that different reviewers may place different subjective weight on “training cost vs inference cost.” However, we kindly ask that the final decision be based on:
>
> - The **stated scope** of the paper (training-time paradigm under fixed architecture);
> - The **full body of evidence** already in the submission (main text + appendix + rebuttal);
> - And reviews that directly engage with those, especially the technically detailed feedback from UEBC.
>
> ---
>
> ### 4. Contribution and value of RepL
>
> Summarizing briefly:
>
> - RepL is, to our knowledge, the **first systematic formulation of Replacement Learning** as a general training paradigm across CNNs, ViTs, and Transformers, with:
>   - a concrete operator ledger for what is removed/added;
>   - a clean theoretical analysis (non-expansive composition, gradient-bias bounds);
>   - targeted span/bias diagnostics that probe the behavior of the replacement operators.
>
> - Experimentally, we provide a **broad and carefully controlled evaluation**:
>   - Multiple architectures (ResNets, ViTs, Transformer-LM);
>   - Multiple tasks (classification, detection, segmentation, language modeling);
>   - Both from-scratch training and pre-trained fine-tuning;
>   - Training-time and inference-time metrics;
>   - Comparisons and combinations with standard training-efficiency baselines (Stochastic Depth, Checkpointing, Skip-Attention).
>
> - Practically, RepL offers a **simple, architecture-agnostic recipe** that any practitioner can apply to:
>   - Train or fine-tune standard backbones under tighter GPU budgets;
>   - Enjoy modest but real inference gains;
>   - And still leverage the full ecosystem of deployment-oriented compression methods afterwards.
>
> We believe this combination of **conceptual novelty**, **breadth of evidence**, and **practical usefulness** meets the bar for an ICLR contribution.
>
> ---
>
> ### 5. Closing
>
> In summary, we respectfully ask that RepL be evaluated as what it is intended and clearly presented to be: **a general training-time efficiency paradigm**, not a replacement for pruning/NAS/distillation. Within that scope, and given the extensive empirical and theoretical support in the paper and appendix, we hope you will find that RepL makes a meaningful and timely contribution to the community, and we would be honored to present it at ICLR.
>
> Thank you very much for your time and consideration.

---

### Note · Program_Chairs · 2026-01-17
**Submission Desk Rejected by Program Chairs**

The following references in this submission do not refer to real documents and/or have major errors in bibliographic information:

 Quoc Tran-Dinh, Benjamin Wild, Stephen Richardson, and Dmitriy Drusvyatskiy. Convergence of Adam and AdamW optimizers. arXiv preprint arXiv:2102.11090, 2021. URL https://arxiv. org/abs/2102.1109